# Champion-level drone racing using deep reinforcement learning

Elia Kaufmann[1✉], Leonard Bauersfeld[1], Antonio Loquercio[1], Matthias Müller[2], Vladlen Koltun[3] & Davide Scaramuzza[1]

First-person view (FPV) drone racing is a televised sport in which professional competitors pilot high-speed aircraft through a 3D circuit. Each pilot sees the environment from the perspective of their drone by means of video streamed from an onboard camera. Reaching the level of professional pilots with an autonomous drone is challenging because the robot needs to fly at its physical limits while estimating its speed and location in the circuit exclusively from onboard sensors[1]. Here we introduce Swift, an autonomous system that can race physical vehicles at the level of the human world champions. The system combines deep reinforcement learning (RL) in simulation with data collected in the physical world. Swift competed against three human champions, including the world champions of two international leagues, in real-world head-to-head races. Swift won several races against each of the human champions and demonstrated the fastest recorded race time. This work represents a milestone for mobile robotics and machine intelligence[2], which may inspire the deployment of hybrid learning-based solutions in other physical systems.

Deep RL[3] has enabled some recent advances in artificial intelligence. Policies trained with deep RL have outperformed humans in complex competitive games, including Atari[4–6], Go[5,7–9], chess[5,9], StarCraft[10], Dota 2 (ref. 11) and Gran Turismo[12,13]. These impressive demonstrations of the capabilities of machine intelligence have primarily been limited to simulation and board-game environments, which support policy search in an exact replica of the testing conditions. Overcoming this limitation and demonstrating champion-level performance in physical competitions is a long-standing problem in autonomous mobile robotics and artificial intelligence[14–16].

FPV drone racing is a televised sport in which highly trained human pilots push aerial vehicles to their physical limits in high-speed agile manoeuvres (Fig. 1a). The vehicles used in FPV racing are quadcopters, which are among the most agile machines ever built (Fig. 1b). During a race, the vehicles exert forces that surpass their own weight by a factor of five or more, reaching speeds of more than 100 km h⁻¹ and accelerations several times that of gravity, even in confined spaces. Each vehicle is remotely controlled by a human pilot who wears a headset showing a video stream from an onboard camera, creating an immersive 'first-person-view' experience (Fig. 1c).

Attempts to create autonomous systems that reach the performance of human pilots date back to the first autonomous drone racing competition in 2016 (ref. 17). A series of innovations followed, including the use of deep networks to identify the next gate location[18–20], transfer of racing policies from simulation to reality[21,22] and accounting for uncertainty in perception[23,24]. The 2019 AlphaPilot autonomous drone racing competition showcased some of the best research in the field[25]. However, the first two teams still took almost twice as long as a professional human pilot to complete the track[26,27]. More recently, autonomous systems have begun to reach expert human performance[28–30]. However, these works rely on near-perfect state estimation provided

by an external motion-capture system. This makes the comparison with human pilots unfair, as humans only have access to onboard observations from the drone.

In this article, we describe Swift, an autonomous system that can race a quadrotor at the level of human world champions using only onboard sensors and computation. Swift consists of two key modules: (1) a perception system that translates high-dimensional visual and inertial information into a low-dimensional representation and (2) a control policy that ingests the low-dimensional representation produced by the perception system and produces control commands.

The control policy is represented by a feedforward neural network and is trained in simulation using model-free on-policy deep RL[31]. To bridge discrepancies in sensing and dynamics between simulation and the physical world, we make use of non-parametric empirical noise models estimated from data collected on the physical system. These empirical noise models have proved to be instrumental for successful transfer of the control policy from simulation to reality.

We evaluate Swift on a physical track designed by a professional drone-racing pilot (Fig. 1a). The track comprises seven square gates arranged in a volume of 30 × 30 × 8 m, forming a lap of 75 m in length. Swift raced this track against three human champions: Alex Vanover, the 2019 Drone Racing League world champion, Thomas Bitmatta, two-time MultiGP International Open World Cup champion, and Marvin Schaepper, three-time Swiss national champion. The quadrotors used by Swift and by the human pilots have the same weight, shape and propulsion. They are similar to drones used in international competitions.

The human pilots were given one week of practice on the race track. After this week of practice, each pilot competed against Swift in several head-to-head races (Fig. 1a,b). In each head-to-head race, two drones

[1]Robotics and Perception Group, University of Zurich, Zürich, Switzerland. [2]Intel Labs, Munich, Germany. [3]Intel Labs, Jackson, WY, USA. ✉e-mail: ekaufmann@ifi.uzh.ch

**a** Drone racing: human versus autonomous

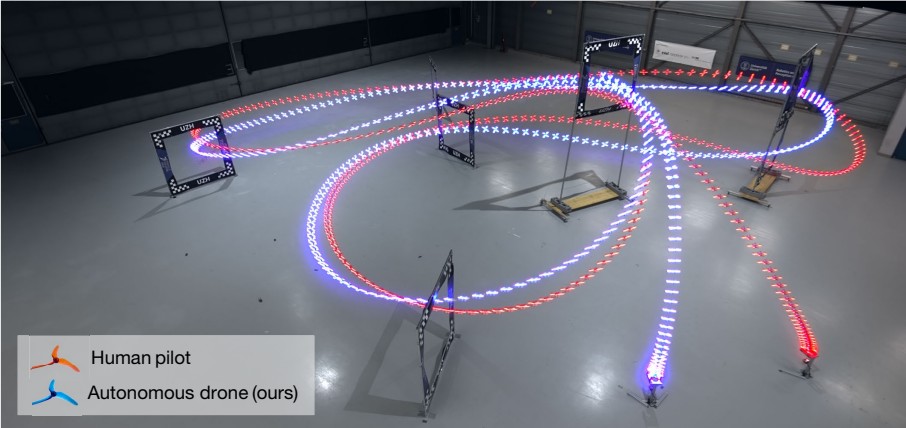

**b** Head-to-head competition

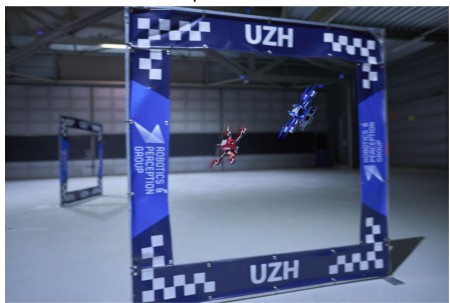

**c** Human champions

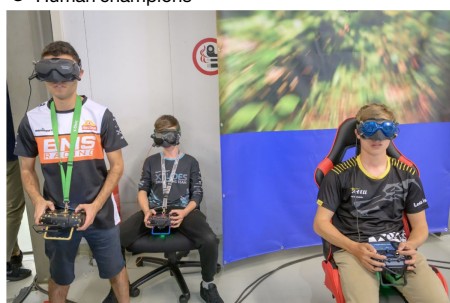

**Fig. 1 | Drone racing. a**, Swift (blue) races head-to-head against Alex Vanover, the 2019 Drone Racing League world champion (red). The track comprises seven square gates that must be passed in order in each lap. To win a race, a competitor has to complete three consecutive laps before its opponent. **b**, A close-up view of Swift, illuminated with blue LEDs, and a human-piloted drone, illuminated with red LEDs. The autonomous drones used in this work rely only on onboard sensory measurements, with no support from external infrastructure, such as motion-capture systems. **c**, From left to right: Thomas Bitmatta, Marvin Schaepper and Alex Vanover racing their drones through the track. Each pilot wears a headset that shows a video stream transmitted in real time from a camera aboard their aircraft. The headsets provide an immersive 'first-person-view' experience. **c**, Photo by Regina Sablotny.

(one controlled by a human pilot and one controlled by Swift) start from a podium. The race is set off by an acoustic signal. The first vehicle that completes three full laps through the track, passing all gates in the correct order in each lap, wins the race.

Swift won several races against each of the human pilots and achieved the fastest race time recorded during the events. Our work marks the first time, to our knowledge, that an autonomous mobile robot achieved world-champion-level performance in a real-world competitive sport.

## The Swift system

Swift uses a combination of learning-based and traditional algorithms to map onboard sensory readings to control commands. This mapping comprises two parts: (1) an observation policy, which distils high-dimensional visual and inertial information into a task-specific low-dimensional encoding, and (2) a control policy that transforms the encoding into commands for the drone. A schematic overview of the system is shown in Fig. 2.

The observation policy consists of a visual–inertial estimator[32,33] that operates together with a gate detector[26], which is a convolutional neural network that detects the racing gates in the onboard images. Detected gates are then used to estimate the global position and orientation of the drone along the race track. This is done by a camera-resectioning algorithm[34] in combination with a map of the track. The estimate of the global pose obtained from the gate detector is then combined with the estimate from the visual–inertial estimator by means of a Kalman filter, resulting in a more accurate representation of the robot's state. The control policy, represented by a two-layer perceptron, maps the output of the Kalman filter to control commands for the aircraft. The policy is trained using on-policy model-free deep RL[31] in simulation. During training, the policy maximizes a reward that combines progress towards the next racing gate[35] with a perception objective that rewards keeping the next gate in the field of view of the camera. Seeing the next gate is rewarded because it increases the accuracy of pose estimation.

Optimizing a policy purely in simulation yields poor performance on physical hardware if the discrepancies between simulation and reality are not mitigated. The discrepancies are caused primarily by two factors: (1) the difference between simulated and real dynamics and (2) the noisy estimation of the robot's state by the observation policy when provided with real sensory data. We mitigate these discrepancies by collecting a small amount of data in the real world and using this data to increase the realism of the simulator.

Specifically, we record onboard sensory observations from the robot together with highly accurate pose estimates from a motion-capture system while the drone is racing through the track. During this data-collection phase, the robot is controlled by a policy trained in simulation that operates on the pose estimates provided by the motion-capture system. The recorded data allow to identify the characteristic failure modes of perception and dynamics observed through the race track. These intricacies of failing perception and unmodelled dynamics are dependent on the environment, platform, track and sensors. The perception and dynamics residuals are modelled using Gaussian processes[36] and k-nearest-neighbour regression, respectively. The motivation behind this choice is that we empirically

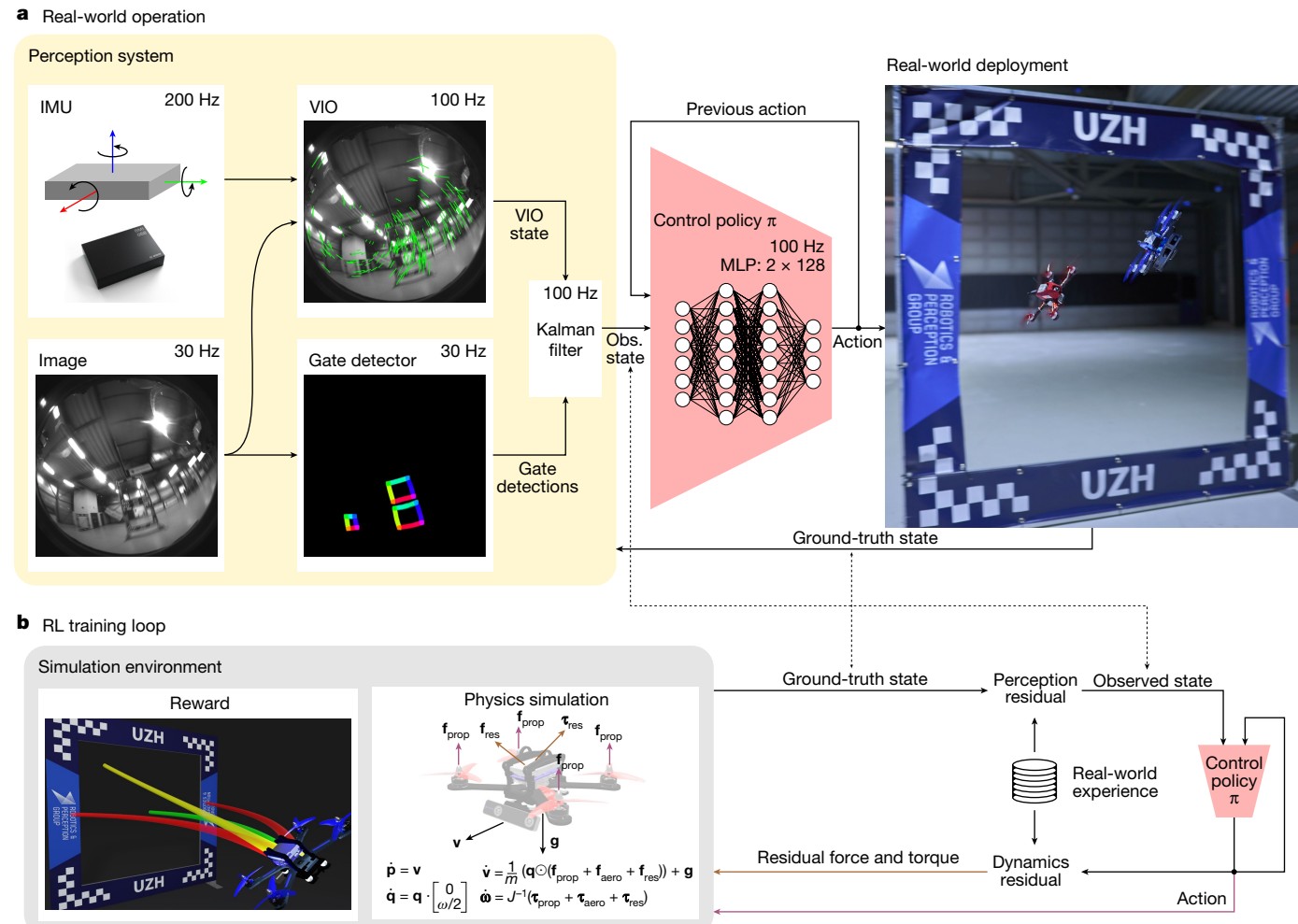

**a** Real-world operation

**b** RL training loop

**Fig. 2 | The Swift system.** Swift consists of two key modules: a perception system that translates visual and inertial information into a low-dimensional state observation and a control policy that maps this state observation to control commands. Control commands specify desired collective thrust and body rates, the same control modality that the human pilots use. **a**, The perception system consists of a VIO module that computes a metric estimate of the drone state from camera images and high-frequency measurements obtained by an inertial measurement unit (IMU). The VIO estimate is coupled with a neural network that detects the corners of racing gates in the image

stream. The corner detections are mapped to a 3D pose and fused with the VIO estimate using a Kalman filter. **b**, We use model-free on-policy deep RL to train the control policy in simulation. During training, the policy maximizes a reward that combines progress towards the centre of the next racing gate with a perception objective to keep the next gate in the field of view of the camera. To transfer the racing policy from simulation to the physical world, we augment the simulation with data-driven residual models of the vehicle's perception and dynamics. These residual models are identified from real-world experience collected on the race track. MLP, multilayer perceptron.

found perception residuals to be stochastic and dynamics residuals to be largely deterministic (Extended Data Fig. 1). These residual models are integrated into the simulation and the racing policy is fine-tuned in this augmented simulation. This approach is related to the empirical actuator models used for simulation-to-reality transfer in ref. 37 but further incorporates empirical modelling of the perception system and also accounts for the stochasticity in the estimate of the platform state.

We ablate each component of Swift in controlled experiments reported in the extended data. Also, we compare against recent work that tackles the task of autonomous drone racing with traditional methods, including trajectory planning and model predictive control (MPC). Although such approaches achieve comparable or even superior performance to our approach in idealized conditions, such as simplified dynamics and perfect knowledge of the robot's state, their performance collapses when their assumptions are violated. We find that approaches that rely on precomputed paths[28,29] are particularly sensitive to noisy perception and dynamics. No traditional method has

achieved competitive lap times compared with Swift or human world champions, even when provided with highly accurate state estimation from a motion-capture system. Detailed analysis is provided in the extended data.

## Results

The drone races take place on a track designed by an external world-class FPV pilot. The track features characteristic and challenging manoeuvres, such as a Split-S (Figs. 1a (top-right corner) and 4d). Pilots are allowed to continue racing even after a crash, provided their vehicle is still able to fly. If both drones crash and cannot complete the track, the drone that proceeded farther along the track wins.

As shown in Fig. 3b, Swift wins 5 out of 9 races against A. Vanover, 4 out of 7 races against T. Bitmatta and 6 out of 9 races against M. Schaepper. Out of the 10 losses recorded for Swift, 40% were because of a collision with the opponent, 40% because of collision with a gate and 20% because of the drone being slower than the human pilot. Overall, Swift

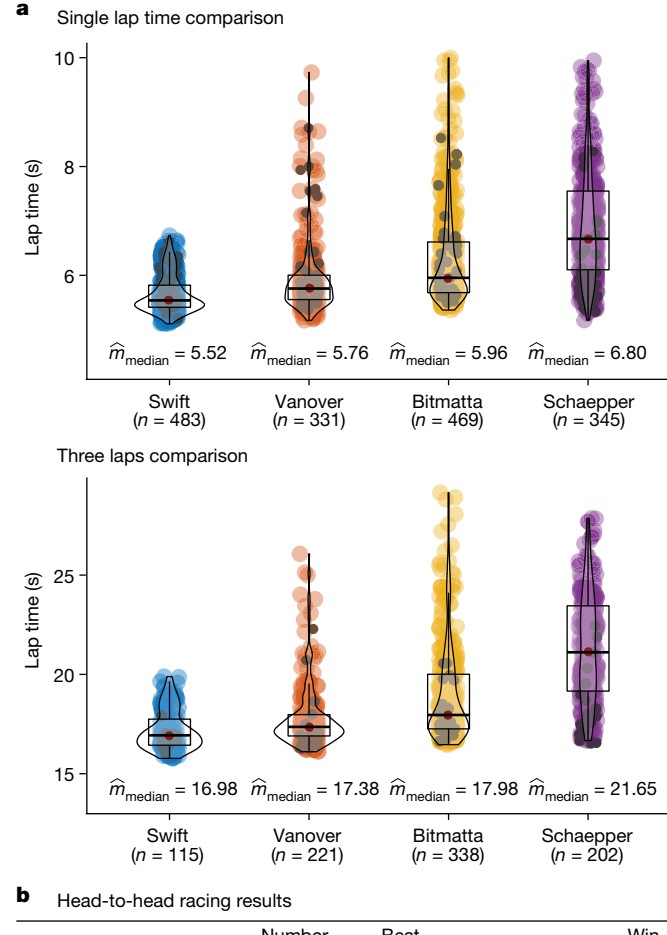

**a**   Single lap time comparison

Lap time (s)

|  | Swift (n = 483) | Vanover (n = 331) | Bitmatta (n = 469) | Schaepper (n = 345) |
|---|---|---|---|---|
| $\widehat{m}_{median}$ | 5.52 | 5.76 | 5.96 | 6.80 |

Three laps comparison

Lap time (s)

|  | Swift (n = 115) | Vanover (n = 221) | Bitmatta (n = 338) | Schaepper (n = 202) |
|---|---|---|---|---|
| $\widehat{m}_{median}$ | 16.98 | 17.38 | 17.98 | 21.65 |

**b**   Head-to-head racing results

|  | Number of races | Best time-to-finish | Wins | Losses | Win ratio |
|---|---|---|---|---|---|
| A. Vanover versus Swift | 9 | 17.956 s | 4 | 5 | 0.44 |
| T. Bitmatta versus Swift | 7 | 18.746 s | 3 | 4 | 0.43 |
| M. Schaepper versus Swift | 9 | 21.160 s | 3 | 6 | 0.33 |
| Swift versus human pilots | 25 | 17.465 s | 15 | 10 | 0.60 |

**Fig. 3 | Results. a**, Lap-time results. We compare Swift against the human pilots in time-trial races. Lap times indicate best single lap times and best average times achieved in a heat of three consecutive laps. The reported statistics are computed over a dataset recorded during one week on the race track, which corresponds to 483 (115) data points for Swift, 331 (221) for A. Vanover, 469 (338) for T. Bitmatta and 345 (202) for M. Schaepper. The first number is the number of single laps and the second is the number of three consecutive laps. The dark points in each distribution correspond to laps flown in race conditions. **b**, Head-to-head results. We report the number of head-to-head races flown by each pilot, the number of wins and losses, as well as the win ratio.

wins most races against each human pilot. Swift also achieves the fastest race time recorded, with a lead of half a second over the best time clocked by a human pilot (A. Vanover).

Figure 4 and Extended Data Table 1d provide an analysis of the fastest lap flown by Swift and each human pilot. Although Swift is globally faster than all human pilots, it is not faster on all individual segments of the track (Extended Data Table 1). Swift is consistently faster at the start and in tight turns such as the split S. At the start, Swift has a lower reaction time, taking off from the podium, on average, 120 ms before human pilots. Also, it accelerates faster and reaches higher speeds going into the first gate (Extended Data Table 1d, segment 1). In sharp turns, as shown in Fig. 4c,d, Swift finds tighter manoeuvres. One hypothesis is that Swift optimizes trajectories on a longer timescale than human pilots. It is known that model-free RL can optimize

long-term rewards through a value function[38]. Conversely, human pilots plan their motion on a shorter timescale, up to one gate into the future[39]. This is apparent, for example in the split S (Fig. 4b,d), for which human pilots are faster in the beginning and at the end of the manoeuvre, but slower overall (Extended Data Table 1d, segment 3). Also, human pilots orient the aircraft to face the next gate earlier than Swift does (Fig. 4c,d). We propose that human pilots are accustomed to keeping the upcoming gate in view, whereas Swift has learned to execute some manoeuvres while relying on other cues, such as inertial data and visual odometry against features in the surrounding environments. Overall, averaged over the entire track, the autonomous drone achieves the highest average speed, finds the shortest racing line and manages to maintain the aircraft closer to its actuation limits throughout the race, as indicated by the average thrust and power drawn (Extended Data Table 1d).

We also compare the performance of Swift and the human champions in time trials (Fig. 3a). In a time trial, a single pilot races the track, with the number of laps left to the discretion of the pilot. We accumulate time-trial data from the practice week and the races, including training runs (Fig. 3a, coloured) and laps flown in race conditions (Fig. 3a, black). For each contestant, we use more than 300 laps for computing statistics. The autonomous drone more consistently pushes for fast lap times, exhibiting lower mean and variance. Conversely, human pilots decide whether to push for speed on a lap-by-lap basis, yielding higher mean and variance in lap times, both during training and in the races. The ability to adapt the flight strategy allows human pilots to maintain a slower pace if they identify that they have a clear lead, so as to reduce the risk of a crash. The autonomous drone is unaware of its opponent and pushes for fastest expected completion time no matter what, potentially risking too much when in the lead and too little when trailing behind[40].

## Discussion

FPV drone racing requires real-time decision-making based on noisy and incomplete sensory input from the physical environment. We have presented an autonomous physical system that achieves champion-level performance in this sport, reaching—and at times exceeding—the performance of human world champions. Our system has certain structural advantages over the human pilots. First, it makes use of inertial data from an onboard inertial measurement unit[32]. This is akin to the human vestibular system[41], which is not used by the human pilots because they are not physically in the aircraft and do not feel the accelerations acting on it. Second, our system benefits from lower sensorimotor latency (40 ms for Swift versus an average of 220 ms for expert human pilots[39]). On the other hand, the limited refresh rate of the camera used by Swift (30 Hz) can be considered a structural advantage for human pilots, whose cameras' refresh rate is four times as fast (120 Hz), improving their reaction time[42].

Human pilots are impressively robust: they can crash at full speed, and—if the hardware still functions—carry on flying and complete the track. Swift was not trained to recover after a crash. Human pilots are also robust to changes in environmental conditions, such as illumination, which can markedly alter the appearance of the track. By contrast, Swift's perception system assumes that the appearance of the environment is consistent with what was observed during training. If this assumption fails, the system can fail. Robustness to appearance changes can be provided by training the gate detector and the residual observation model in a diverse set of conditions. Addressing these limitations could enable applying the presented approach in autonomous drone racing competitions in which access to the environment and the drone is limited[25].

Notwithstanding the remaining limitations and the work ahead, the attainment by an autonomous mobile robot of world-champion-level performance in a popular physical sport is a milestone for robotics

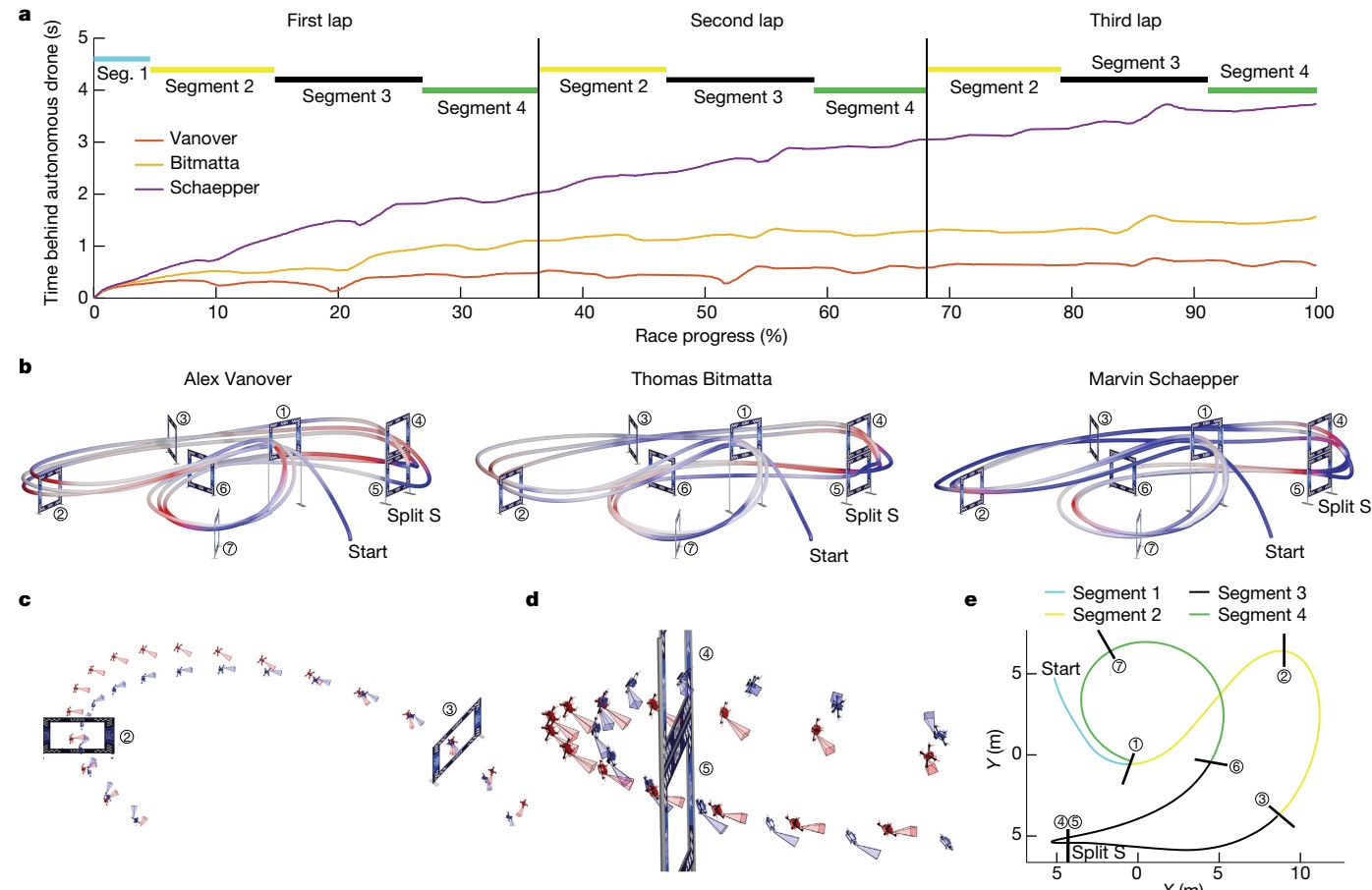

**Fig. 4 | Analysis. a**, Comparison of the fastest race of each pilot, illustrated by the time behind Swift. The time difference from the autonomous drone is computed as the time since it passed the same position on the track. Although Swift is globally faster than all human pilots, it is not necessarily faster on all individual segments of the track. **b**, Visualization of where the human pilots are faster (red) and slower (blue) compared with the autonomous drone. Swift is consistently faster at the start and in tight turns, such as the split S. **c**, Analysis of the manoeuvre after gate 2. Swift in blue, Vanover in red. Swift gains time against human pilots in this segment as it executes a tighter turn while maintaining comparable speed. **d**, Analysis of the split S manoeuvre. Swift in blue, Vanover in red. The split S is the most challenging segment in the race track, requiring a carefully coordinated roll and pitch motion that yields a descending half-loop through the two gates. Swift gains time against human pilots on this segment as it executes a tighter turn with less overshoot. **e**, Illustration of track segments used for analysis. Segment 1 is traversed once at the start, whereas segments 2–4 are traversed in each lap (three times over the course of a race).

and machine intelligence. This work may inspire the deployment of hybrid learning-based solutions in other physical systems, such as autonomous ground vehicles, aircraft and personal robots, across a broad range of applications.

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

## Methods

### Quadrotor simulation

**Quadrotor dynamics.** To enable large-scale training, we use a high-fidelity simulation of the quadrotor dynamics. This section briefly explains the simulation. The dynamics of the vehicle can be written as

$$
\dot{\mathbf{x}} = \begin{bmatrix} \dot{\mathbf{p}}_{\mathcal{WB}} \\ \dot{\mathbf{q}}_{\mathcal{WB}} \\ \dot{\mathbf{v}}_{\mathcal{W}} \\ \dot{\boldsymbol{\omega}}_{\mathcal{B}} \\ \dot{\boldsymbol{\Omega}} \end{bmatrix} = \begin{bmatrix} \mathbf{v}_{\mathcal{W}} \\ \mathbf{q}_{\mathcal{WB}} \cdot \begin{bmatrix} 0 \\ \boldsymbol{\omega}_{\mathcal{B}}/2 \end{bmatrix} \\ \frac{1}{m}(\mathbf{q}_{\mathcal{WB}} \odot (\mathbf{f}_{\text{prop}} + \mathbf{f}_{\text{aero}})) + \mathbf{g}_{\mathcal{W}} \\ J^{-1}(\boldsymbol{\tau}_{\text{prop}} + \boldsymbol{\tau}_{\text{mot}} + \boldsymbol{\tau}_{\text{aero}} + \boldsymbol{\tau}_{\text{iner}}) \\ \frac{1}{k_{\text{mot}}}(\boldsymbol{\Omega}_{\text{ss}} - \boldsymbol{\Omega}) \end{bmatrix}, \tag{1}
$$

in which $\odot$ represents quaternion rotation, $\mathbf{p}_{\mathcal{WB}}$, $\mathbf{q}_{\mathcal{WB}}$, $\mathbf{v}_{\mathcal{W}}$ and $\boldsymbol{\omega}_{\mathcal{B}}$ denote the position, attitude quaternion, inertial velocity and body rates of the quadcopter, respectively. The motor time constant is $k_{\text{mot}}$ and the motor speeds $\boldsymbol{\Omega}$ and $\boldsymbol{\Omega}_{\text{ss}}$ are the actual and steady-state motor speeds, respectively. The matrix $J$ is the inertia of the quadcopter and $\mathbf{g}_{\mathcal{W}}$ denotes the gravity vector. Two forces act on the quadrotor: the lift force $\mathbf{f}_{\text{prop}}$ generated by the propellers and an aerodynamic force $\mathbf{f}_{\text{aero}}$ that aggregates all other forces, such as aerodynamic drag, dynamic lift and induced drag. The torque is modelled as a sum of four components: the torque generated by the individual propeller thrusts $\boldsymbol{\tau}_{\text{prop}}$, the yaw torque $\boldsymbol{\tau}_{\text{mot}}$ generated by a change in motor speed, an aerodynamic torque $\boldsymbol{\tau}_{\text{aero}}$ that accounts for various aerodynamic effects such as blade flapping and an inertial term $\boldsymbol{\tau}_{\text{iner}}$. The individual components are given as

$$
\mathbf{f}_{\text{prop}} = \sum_i \mathbf{f}_i, \ \boldsymbol{\tau}_{\text{prop}} = \sum_i \boldsymbol{\tau}_i + \mathbf{r}_{\text{P},i} \times \mathbf{f}_i, \tag{2}
$$

$$
\boldsymbol{\tau}_{\text{mot}} = J_{\text{m+p}} \sum_i \boldsymbol{\zeta}_i \dot{\Omega}_i, \ \boldsymbol{\tau}_{\text{iner}} = -\boldsymbol{\omega}_{\mathcal{B}} \times \mathbf{J}\boldsymbol{\omega}_{\mathcal{B}} \tag{3}
$$

in which $\mathbf{r}_{\text{P},i}$ is the location of propeller $i$, expressed in the body frame, and $\mathbf{f}_i$ and $\boldsymbol{\tau}_i$ are the forces and torques, respectively, generated by the $i$th propeller. The axis of rotation of the $i$th motor is denoted by $\boldsymbol{\zeta}_i$, the combined inertia of the motor and propeller is $J_{\text{m+p}}$ and the derivative of the $i$th motor speed is $\dot{\Omega}_i$. The individual propellers are modelled using a commonly used quadratic model, which assumes that the lift force and drag torque are proportional to the square of the propeller speed $\Omega_i$:

$$
\mathbf{f}_i(\Omega_i) = \begin{bmatrix} 0 & 0 & c_l \cdot \Omega_i^2 \end{bmatrix}^\top, \quad \boldsymbol{\tau}_i(\Omega_i) = \begin{bmatrix} 0 & 0 & c_d \cdot \Omega_i^2 \end{bmatrix}^\top \tag{4}
$$

in which $c_l$ and $c_d$ denote the propeller lift and drag coefficients, respectively.

**Aerodynamic forces and torques.** The aerodynamic forces and torques are difficult to model with a first-principles approach. We thus use a data-driven model[43]. To maintain the low computational complexity required for large-scale RL training, a grey-box polynomial model is used rather than a neural network. The aerodynamic effects are assumed to primarily depend on the velocity $\mathbf{v}_{\mathcal{B}}$ (in the body frame) and the average squared motor speed $\overline{\Omega}^2$. The aerodynamic forces $f_x, f_y$ and $f_z$ and torques $\tau_x$, $\tau_y$ and $\tau_z$ are estimated in the body frame. The quantities $v_x$, $v_y$ and $v_z$ denote the three axial velocity components (in the body frame) and $v_{xy}$ denotes the speed in the $(x, y)$ plane of the quadrotor. On the basis of insights from the underlying physical processes, linear and quadratic combinations of the individual terms are selected. For readability, the coefficients multiplying each summand have been omitted:

$$
f_x \sim v_x + v_x|v_x| + \overline{\Omega}^2 + v_x \ \overline{\Omega}^2
$$

$$
f_y \sim v_y + v_y|v_y| + \overline{\Omega}^2 + v_y \ \overline{\Omega}^2
$$

$$
f_z \sim v_z + v_z|v_z| + v_{xy} + v_{xy}^2 + v_{xy} \ \overline{\Omega}^2 + v_z \ \overline{\Omega}^2 + v_{xy} v_z \ \overline{\Omega}^2
$$

$$
\tau_x \sim v_y + v_y|v_y| + \overline{\Omega}^2 + v_y \ \overline{\Omega}^2 + v_y|v_y| \ \overline{\Omega}^2
$$

$$
\tau_y \sim v_x + v_x|v_x| + \overline{\Omega}^2 + v_x \ \overline{\Omega}^2 + v_x|v_x| \ \overline{\Omega}^2
$$

$$
\tau_z \sim v_x + v_y
$$

The respective coefficients are then identified from real-world flight data, in which motion capture is used to provide ground-truth forces and torque measurements. We use data from the race track, allowing the dynamics model to fit the track. This is akin to the human pilots' training for days or weeks before the race on the specific track that they will be racing. In our case, the human pilots are given a week of practice on the same track ahead of the competition.

**Betaflight low-level controller.** To control the quadrotor, the neural network outputs collective thrust and body rates. This control signal is known to combine high agility with good robustness to simulation-to-reality transfer[44]. The predicted collective thrust and body rates are then processed by an onboard low-level controller that computes individual motor commands, which are subsequently translated into analogue voltage signals through an electronic speed controller (ESC) that controls the motors. On the physical vehicle, this low-level proportional–integral–derivative (PID) controller and ESC are implemented using the open-source Betaflight and BLHeli32 firmware[45]. In simulation, we use an accurate model of both the low-level controller and the motor speed controller.

Because the Betaflight PID controller has been optimized for human-piloted flight, it exhibits some peculiarities, which the simulation correctly captures: the reference for the D-term is constantly zero (pure damping), the I-term gets reset when the throttle is cut and, under motor thrust saturation, the body rate control is assigned priority (proportional downscaling of all motor signals to avoid saturation). The gains of the controller used for simulation have been identified from the detailed logs of the Betaflight controller's internal states. The simulation can predict the individual motor commands with less than 1% error.

**Battery model and ESC.** The low-level controller converts the individual motor commands into a pulse-width modulation (PWM) signal and sends it to the ESC, which controls the motors. Because the ESC does not perform closed-loop control of the motor speeds, the steady-state motor speed $\Omega_{i,\text{ss}}$ for a given PWM motor command $\text{cmd}_i$ is a function of the battery voltage. Our simulation thus models the battery voltage using a grey-box battery model[46] that simulates the voltage based on instantaneous power consumption $P_{\text{mot}}$:

$$
P_{\text{mot}} = \frac{c_d \Omega^3}{\eta} \tag{5}
$$

The battery model[46] then simulates the battery voltage based on this power demand. Given the battery voltage $U_{\text{bat}}$ and the individual motor command $u_{\text{cmd},i}$, we use the mapping (again omitting the coefficients multiplying each summand)

$$
\Omega_{i,\text{ss}} \sim 1 + U_{\text{bat}} + \sqrt{u_{\text{cmd},i}} + u_{\text{cmd},i} + U_{\text{bat}}\sqrt{u_{\text{cmd},i}} \tag{6}
$$

to calculate the corresponding steady-state motor speed $\Omega_{i,\text{ss}}$ required for the dynamics simulation in equation (1). The coefficients have been

identified from Betaflight logs containing measurements of all involved quantities. Together with the model of the low-level controller, this enables the simulator to correctly translate an action in the form of collective thrust and body rates to desired motor speeds $\Omega_{ss}$ in equation (1).

## Policy training

We train deep neural control policies that directly map observations $\mathbf{o}_t$ in the form of platform state and next gate observation to control actions $\mathbf{u}_t$ in the form of mass-normalized collective thrust and body rates[44]. The control policies are trained using model-free RL in simulation.

**Training algorithm.** Training is performed using proximal policy optimization[31]. This actor-critic approach requires jointly optimizing two neural networks during training: the policy network, which maps observations to actions, and the value network, which serves as the 'critic' and evaluates actions taken by the policy. After training, only the policy network is deployed on the robot.

**Observations, actions and rewards.** An observation $\mathbf{o}_t \in \mathbb{R}^{31}$ obtained from the environment at time $t$ consists of: (1) an estimate of the current robot state; (2) the relative pose of the next gate to be passed on the track layout; and (3) the action applied in the previous step. Specifically, the estimate of the robot state contains the position of the platform, its velocity and attitude represented by a rotation matrix, resulting in a vector in $\mathbb{R}^{15}$. Although the simulation uses quaternions internally, we use a rotation matrix to represent attitude to avoid ambiguities[47]. The relative pose of the next gate is encoded by providing the relative position of the four gate corners with respect to the vehicle, resulting in a vector in $\mathbb{R}^{12}$. All observations are normalized before being passed to the network. Because the value network is only used during training time, it can access privileged information about the environment that is not accessible to the policy[48]. This privileged information is concatenated with other inputs to the policy network and contains the exact position, orientation and velocity of the robot.

For each observation $\mathbf{o}_t$, the policy network produces an action $\mathbf{a}_t \in \mathbb{R}^4$ in the form of desired mass-normalized collective thrust and body rates.

We use a dense shaped reward formulation to learn the task of perception-aware autonomous drone racing. The reward $r_t$ at time step $t$ is given by

$$r_t = r_t^{\text{prog}} + r_t^{\text{perc}} + r_t^{\text{cmd}} - r_t^{\text{crash}} \tag{7}$$

in which $r^{\text{prog}}$ rewards progress towards the next gate[35], $r^{\text{perc}}$ encodes perception awareness by adjusting the attitude of the vehicle such that the optical axis of the camera points towards the centre of the next gate, $r^{\text{cmd}}$ rewards smooth actions and $r^{\text{crash}}$ is a binary penalty that is only active when colliding with a gate or when the platform leaves a predefined bounding box. If $r^{\text{crash}}$ is triggered, the training episode ends. Specifically, the reward terms are

$$r_t^{\text{prog}} = \lambda_1 [d_{t-1}^{\text{Gate}} - d_t^{\text{Gate}}] \tag{8}$$
$$r_t^{\text{perc}} = \lambda_2 \exp[\lambda_3 \cdot \delta_{\text{cam}}^4]$$

$$r_t^{\text{cmd}} = \lambda_4 \mathbf{a}_t^\omega + \lambda_5 \|\mathbf{a}_t - \mathbf{a}_{t-1}\|^2 \tag{9}$$

$$r_t^{\text{crash}} = \begin{cases} 5.0, & \text{if } p_z < 0 \text{ or in collision with gate} \\ 0, & \text{otherwise} \end{cases}$$

in which $d_t^{\text{Gate}}$ denotes the distance from the centre of mass of the vehicle to the centre of the next gate at time step $t$, $\delta_{\text{cam}}$ represents the angle between the optical axis of the camera and the centre of the next gate

and $\mathbf{a}_t^\omega$ are the commanded body rates. The hyperparameters $\lambda_1, \ldots, \lambda_5$ balance different terms (Extended Data Table 1a).

**Training details.** Data collection is performed by simulating 100 agents in parallel that interact with the environment in episodes of 1,500 steps. At each environment reset, every agent is initialized at a random gate on the track, with bounded perturbation around a state previously observed when passing this gate. In contrast to previous work[44,49,50], we do not perform randomization of the platform dynamics at training time. Instead, we perform fine-tuning based on real-world data. The training environment is implemented using TensorFlow Agents[51]. The policy network and the value network are both represented by two-layer perceptrons with 128 nodes in each layer and LeakyReLU activations with a negative slope of 0.2. Network parameters are optimized using the Adam optimizer with learning rate $3 \times 10^{-4}$ for both the policy network and the value network.

Policies are trained for a total of $1 \times 10^8$ environment interactions, which takes 50 min on a workstation (i9 12900K, RTX 3090, 32 GB RAM DDR5). Fine-tuning is performed for $2 \times 10^7$ environment interactions.

## Residual model identification

We perform fine-tuning of the original policy based on a small amount of data collected in the real world. Specifically, we collect three full rollouts in the real world, corresponding to approximately 50 s of flight time. We fine-tune the policy by identifying residual observations and residual dynamics, which are then used for training in simulation. During this fine-tuning phase, only the weights of the control policy are updated, whereas the weights of the gate-detection network are kept constant.

**Residual observation model.** Navigating at high speeds results in substantial motion blur, which can lead to a loss of tracked visual features and severe drift in linear odometry estimates. We fine-tune policies with an odometry model that is identified from only a handful of trials recorded in the real world. To model the drift in odometry, we use Gaussian processes[36], as they allow fitting a posterior distribution of odometry perturbations, from which we can sample temporally consistent realizations.

Specifically, the Gaussian process model fits residual position, velocity and attitude as a function of the ground-truth robot state. The observation residuals are identified by comparing the observed visual–inertial odometry (VIO) estimates during a real-world rollout with the ground-truth platform states, which are obtained from an external motion-tracking system.

We treat each dimension of the observation separately, effectively fitting a set of nine 1D Gaussian processes to the observation residuals. We use a mixture of radial basis function kernels

$$\kappa(\mathbf{z}_i, \mathbf{z}_j) = \sigma_f^2 \exp\left(-\frac{1}{2}(\mathbf{z}_i - \mathbf{z}_j)^\top L^{-2}(\mathbf{z}_i - \mathbf{z}_j)\right) + \sigma_n^2 \tag{10}$$

in which $L$ is the diagonal length scale matrix and $\sigma_f$ and $\sigma_n$ represent the data and prior noise variance, respectively, and $\mathbf{z}_i$ and $\mathbf{z}_j$ represent data features. The kernel hyperparameters are optimized by maximizing the log marginal likelihood. After kernel hyperparameter optimization, we sample new realizations from the posterior distribution that are then used during fine-tuning of the policy. Extended Data Fig. 1 illustrates the residual observations in position, velocity and attitude in real-world rollouts, as well as 100 sampled realizations from the Gaussian process model.

**Residual dynamics model.** We use a residual model to complement the simulated robot dynamics[52]. Specifically, we identify residual accelerations as a function of the platform state $\mathbf{s}$ and the commanded mass-normalized collective thrust $c$:

$$\mathbf{a}_{\text{res}} = \text{KNN}(\mathbf{s}, c) \tag{11}$$

We use $k$-nearest neighbour regression with $k = 5$. The size of the dataset used for residual dynamics model identification depends on the track layout and ranges between 800 and 1,000 samples for the track layout used in this work.

## Gate detection

To correct for drift accumulated by the VIO pipeline, the gates are used as distinct landmarks for relative localization. Specifically, gates are detected in the onboard camera view by segmenting gate corners[26]. The greyscale images provided by the Intel RealSense Tracking Camera T265 are used as input images for the gate detector. The architecture of the segmentation network is a six-level U-Net[53] with (8, 16, 16, 16, 16, 16) convolutional filters of size (3, 3, 3, 5, 7, 7) per level and a final extra layer operating on the output of the U-Net containing 12 filters. As the activation function, LeakyReLU with $\alpha = 0.01$ is used. For deployment on the NVIDIA Jetson TX2, the network is ported to TensorRT. To optimize memory footprint and computation time, inference is performed in half-precision mode (FP16) and images are downsampled to size $384 \times 384$ before being fed to the network. One forward pass through the network takes 40 ms on the NVIDIA Jetson TX2.

## VIO drift estimation

The odometry estimates from the VIO pipeline[54] exhibit substantial drift during high-speed flight. We use gate detection to stabilize the pose estimates produced by VIO. The gate detector outputs the coordinates of the corners of all visible gates. A relative pose is first estimated for all predicted gates using infinitesimal plane-based pose estimation (IPPE)[34]. Given this relative pose estimate, each gate observation is assigned to the closest gate in the known track layout, thus yielding a pose estimate for the drone.

Owing to the low frequency of the gate detections and the high quality of the VIO orientation estimate, we only refine the translational components of the VIO measurements. We estimate and correct for the drift of the VIO pipeline using a Kalman filter that estimates the translational drift $\mathbf{p}_d$ (position offset) and its derivative, the drift velocity $\mathbf{v}_d$. The drift correction is performed by subtracting the estimated drift states $\mathbf{p}_d$ and $\mathbf{v}_d$ from the corresponding VIO estimates. The Kalman filter state $\mathbf{x}$ is given by $\mathbf{x} = [\mathbf{p}_d^\top, \mathbf{v}_d^\top]^\top \in \mathbb{R}^6$.

The state $\mathbf{x}$ and covariance $P$ updates are given by:

$$\mathbf{x}_{k+1} = F\mathbf{x}_k, \quad P_{k+1} = FP_kF^\top + Q, \tag{12}$$

$$F = \begin{bmatrix} \mathbb{I}^{3\times3} & \mathrm{d}t\,\mathbb{I}^{3\times3} \\ 0^{3\times3} & \mathbb{I}^{3\times3} \end{bmatrix}, \quad Q = \begin{bmatrix} \sigma_{\text{pos}}\mathbb{I}^{3\times3} & 0^{3\times3} \\ 0^{3\times3} & \sigma_{\text{vel}}\mathbb{I}^{3\times3} \end{bmatrix}. \tag{13}$$

On the basis of measurements, the process noise is set to $\sigma_{\text{pos}} = 0.05$ and $\sigma_{\text{vel}} = 0.1$. The filter state and covariance are initialized to zero. For each measurement $\mathbf{z}_k$ (pose estimate from a gate detection), the predicted VIO drift $\mathbf{x}_k^-$ is corrected to the estimate $\mathbf{x}_k^+$ according to the Kalman filter equations:

$$K_k = P_k^- H_k^\top (H_k P_k^- H_k^\top + R)^{-1},$$
$$\mathbf{x}_k^+ = \mathbf{x}_k^- + K_k(\mathbf{z}_k - H(\mathbf{x}_k^-)), \tag{14}$$
$$P_k^+ = (I - K_k H_k)P_k^-,$$

in which $K_k$ is the Kalman gain, $R$ is the measurement covariance and $H_k$ is the measurement matrix. If several gates have been detected in a single camera frame, all relative pose estimates are stacked and processed in the same Kalman filter update step. The main source of measurement error is the uncertainty in the gate-corner detection of the network. This error in the image plane results in a pose error when IPPE is applied.

We opted for a sampling-based approach to estimate the pose error from the known average gate-corner-detection uncertainty. For each gate, the IPPE algorithm is applied to the nominal gate observation as well as to 20 perturbed gate-corner estimates. The resulting distribution of pose estimates is then used to approximate the measurement covariance $R$ of the gate observation.

## Simulation results

Reaching champion-level performance in autonomous drone racing requires overcoming two challenges: imperfect perception and incomplete models of the system's dynamics. In controlled experiments in simulation, we assess the robustness of our approach to both of these challenges. To this end, we evaluate performance in a racing task when deployed in four different settings. In setting (1), we simulate a simplistic quadrotor model with access to ground-truth state observations. In setting (2), we replace the ground-truth state observations with noisy observations identified from real-world flights. These noisy observations are generated by sampling one realization from the residual observation model and are independent of the perception awareness of the deployed controller. Settings (3) and (4) share the observation models with the previous two settings, respectively, but replace the simplistic dynamics model with more accurate aerodynamical simulation[43]. These four settings allow controlled assessment of the sensitivity of the approach to changes in the dynamics and the observation fidelity.

In all four settings, we benchmark our approach against the following baselines: zero-shot, domain randomization and time-optimal. The zero-shot baseline represents a learning-based racing policy[35] trained using model-free RL that is deployed zero-shot from the training domain to the test domain. The training domain of the policy is equal to experimental setting (1), that is, idealized dynamics and ground-truth observations. Domain randomization extends the learning strategy from the zero-shot baseline by randomizing observations and dynamics properties to increase robustness. The time-optimal baseline uses a precomputed time-optimal trajectory[28] that is tracked using an MPC controller. This approach has shown the best performance in comparison with other model-based methods for time-optimal flight[55,56]. The dynamics model used by the trajectory generation and the MPC controller matches the simulated dynamics of experimental setting (1).

Performance is assessed by evaluating the fastest lap time, the average and minimum observed gate margin of successfully passed gates and the percentage of track successfully completed. The gate margin metric measures the distance between the drone and the closest point on the gate when crossing the gate plane. A high gate margin indicates that the quadrotor passed close to the centre of the gate. Leaving a smaller gate margin can increase speed but can also increase the risk of collision or missing the gate. Any lap that results in a crash is not considered valid.

The results are summarized in Extended Data Table 1c. All approaches manage to successfully complete the task when deployed in idealized dynamics and ground-truth observations, with the time-optimal baseline yielding the lowest lap time. When deployed in settings that feature domain shift, either in the dynamics or the observations, the performance of all baselines collapses and none of the three baselines are able to complete even a single lap. This performance drop is exhibited by both learning-based and traditional approaches. By contrast, our approach, which features empirical models of dynamics and observation noise, succeeds in all deployment settings, with small increases in lap time.

The key feature that enables our approach to succeed across deployment regimes is the use of an empirical model of dynamics and observation noise, estimated from real-world data. A comparison between an approach that has access to such data and approaches that do not is not entirely fair. For that reason, we also benchmark the performance of all baseline approaches when having access to the same real-world data used by our approach. Specifically, we

compare the performance in experimental setting (2), which features the idealized dynamics model but noisy perception. All baseline approaches are provided with the predictions of the same Gaussian process model that we use to characterize observation noise. The results are summarized in Extended Data Table 1b. All baselines benefit from the more realistic observations, yielding higher completion rates. Nevertheless, our approach is the only one that reliably completes the entire track. As well as the predictions of the observation noise model, our approach also takes into account the uncertainty of the model. For an in-depth comparison of the performance of RL versus optimal control in controlled experiments, we refer the reader to ref. 57.

### Fine-tuning for several iterations

We investigate the extent of variations in behaviour across iterations. The findings of our analysis reveal that subsequent fine-tuning operations result in negligible enhancements in performance and alterations in behaviour (Extended Data Fig. 2).

In the following, we provide more details on this investigation. We start by enumerating the fine-tuning steps to provide the necessary notation:

1. Train policy-0 in simulation.
2. Deploy policy-0 in the real world. The policy operates on ground-truth data from a motion-capture system.
3. Identify residuals observed by policy-0 in the real world.
4. Train policy-1 by fine-tuning policy-0 on the identified residuals.
5. Deploy policy-1 in the real world. The policy operates only on onboard sensory measurements.
6. Identify residuals observed by policy-1 in the real world.
7. Train policy-2 by fine-tuning policy-1 on the identified residuals.

We compare the performance of policy-1 and policy-2 in simulation after fine-tuning on their respective residuals. The results are illustrated in Extended Data Fig. 2. We observe that the difference in distance from gate centres, which is a metric for the safety of the policy, is 0.09 ± 0.08 m. Furthermore, the difference in the time taken to complete a single lap is 0.02 ± 0.02 s. Note that this lap-time difference is substantially smaller than the difference between the single-lap completion times of Swift and the human pilots (0.16 s).

### Drone hardware configuration

The quadrotors used by the human pilots and Swift have the same weight, shape and propulsion. The platform design is based on the Agilicious framework[58]. Each vehicle has a weight of 870 g and can produce a maximum static thrust of approximately 35 N, which results in a static thrust-to-weight ratio of 4.1. The base of each platform consists of an Armattan Chameleon 6″ main frame that is equipped with T-Motor Velox 2306 motors and 5″, three-bladed propellers. An NVIDIA Jetson TX2 accompanied by a Connect Tech Quasar carrier board provides the main compute resource for the autonomous drones, featuring a six-core CPU running at 2 GHz and a dedicated GPU with 256 CUDA cores running at 1.3 GHz. Although forward passes of the gate-detection network are performed on the GPU, the racing policy is evaluated on the CPU, with one inference pass taking 8 ms. The autonomous drones carry an Intel RealSense Tracking Camera T265 that provides VIO estimates[59] at 100 Hz that are fed by USB to the NVIDIA Jetson TX2. The human-piloted drones carry neither a Jetson computer nor a RealSense camera and are instead equipped with a corresponding ballast weight. Control commands in the form of collective thrust and body rates produced by the human pilots or Swift are sent to a commercial flight controller, which runs on an STM32 processor operating at 216 MHz. The flight controller is running Betaflight, an open-source flight-control software[45].

### Human pilot impressions

The following quotes convey the impressions of the three human champions who raced against Swift.

**Alex Vanover:**
- These races will be decided at the split S, it is the most challenging part of the track.
- This was the best race! I was so close to the autonomous drone, I could really feel the turbulence when trying to keep up with it.

**Thomas Bitmatta:**
- The possibilities are endless, this is the start of something that could change the whole world. On the flip side, I'm a racer, I don't want anything to be faster than me.
- As you fly faster, you trade off precision for speed.
- It's inspiring to see the potential of what drones are actually capable of. Soon, the AI drone could even be used as a training tool to understand what would be possible.

**Marvin Schaepper:**
- It feels different racing against a machine, because you know that the machine doesn't get tired.

### Research ethics

The study has been conducted in accordance with the Declaration of Helsinki. The study protocol is exempt from review by an ethics committee according to the rules and regulations of the University of Zurich, because no health-related data has been collected. The participants gave their written informed consent before participating in the study.

## Data availability

All (other) data needed to evaluate the conclusions in the paper are present in the paper or the extended data. Motion-capture recordings of the race events with accompanying analysis code can be found in the file 'racing_data.zip' on Zenodo at https://doi.org/10.5281/zenodo.7955278.

## Code availability

Pseudocode for Swift detailing the training process and algorithms can be found in the file 'pseudocode.zip' on Zenodo at https://doi.org/10.5281/zenodo.7955278. To safeguard against potential misuse, the full source code associated with this research will not be made publicly available.

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

**Acknowledgements** The authors thank A. Vanover, T. Bitmatta and M. Schaepper for accepting to race against Swift. The authors also thank C. Pfeiffer, T. Längle and A. Barden for their contributions to the organization of the race events and the drone hardware design. This work was supported by Intel's Embodied AI Lab, the Swiss National Science Foundation (SNSF) through the National Centre of Competence in Research (NCCR) Robotics and the European Research Council (ERC) under grant agreement 864042 (AGILEFLIGHT).

**Author contributions** E.K. formulated the main ideas, implemented the system, performed the experiments and data analysis and wrote the paper. L.B. contributed to the main ideas, the experiments, data analysis, paper writing and designed the graphical illustrations. A.L. formulated the main ideas and contributed to the experimental design, data analysis and paper writing. M.M. contributed to the experimental design, data analysis and paper writing. V.K. contributed to the main ideas, the experimental design, the analysis of experiments and paper writing. D.S. contributed to the main ideas, experimental design, analysis of experiments, paper writing and provided funding.

**Competing interests** The authors declare no competing interests.

**Additional information**
**Correspondence and requests for materials** should be addressed to Elia Kaufmann.

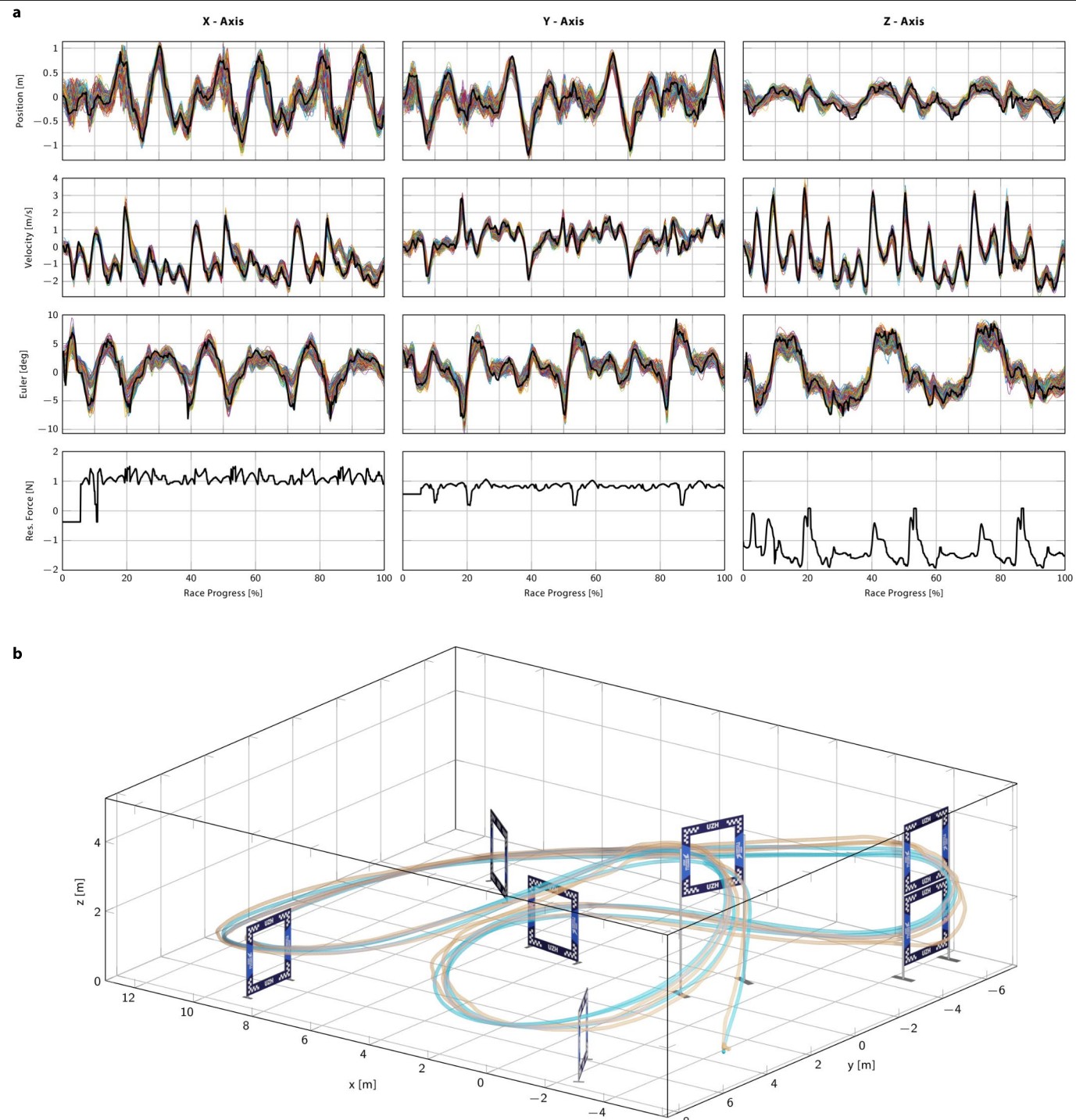

**Extended Data Fig. 1 | Residual models. a**, Visualization of the residual observation model and the residual dynamics model identified from real-world data. Black curves depict the residual observed in the real world and coloured lines show 100 sampled realizations of the residual observation model. Each plot depicts an entire race, that is, three laps. **b**, Predicted residual observation for a simulated rollout. Blue, ground-truth position provided by the simulator; orange, perturbed position generated by the Gaussian process residual.

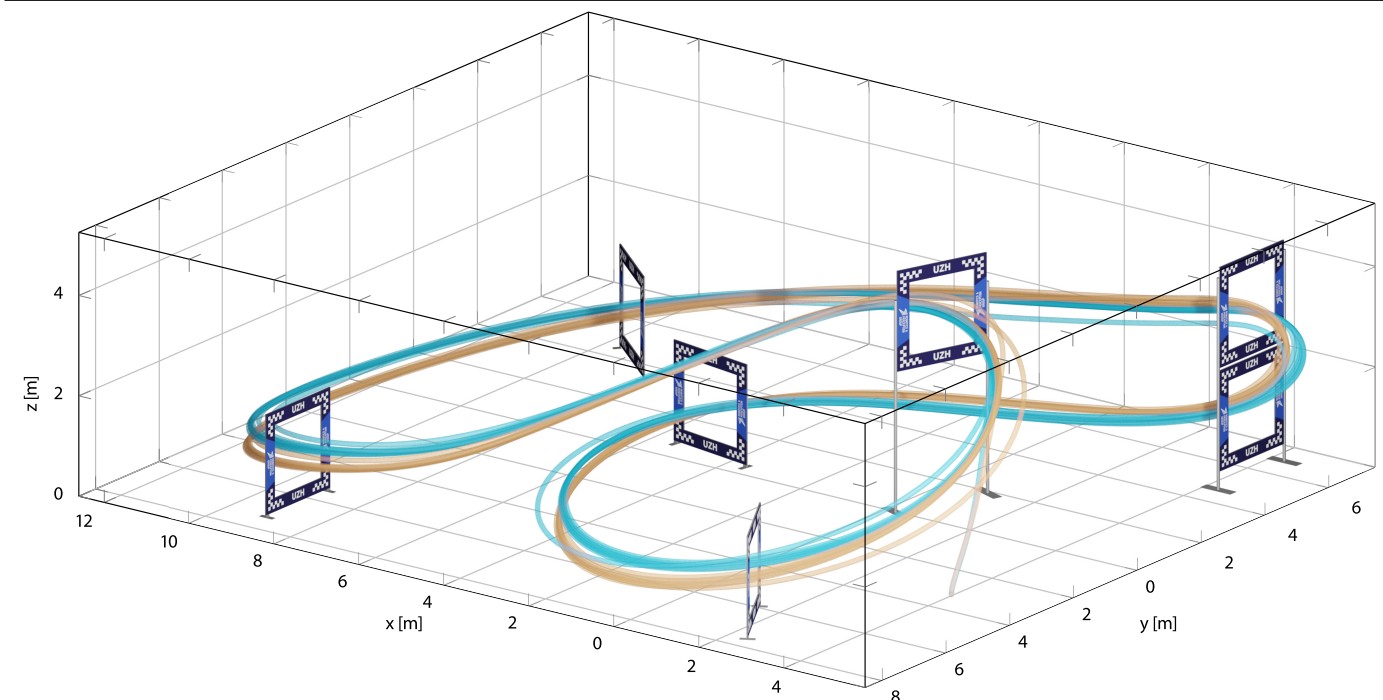

**Extended Data Fig. 2 | Multi-iteration fine-tuning.** Rollout comparison after fine-tuning the policy for one iteration (blue) and two iterations (orange).

**Extended Data Table 1 | Network parameters and detailed metrics show how our Swift system compares with other approaches**

**a**

| Hyperparameter | Value |
|---|---|
| $\gamma$ (discount factor) | 0.99 |
| $\varepsilon$ (importance ratio clipping) | 0.2 |
| $\lambda_1$ | 1.0 |
| $\lambda_2$ | 0.02 |
| $\lambda_3$ | -10.0 |
| $\lambda_4$ | -2e-4 |
| $\lambda_5$ | -1e-4 |

**b**

| Approach | Lap time in seconds | Gate margin in meters | Completion in % |
|---|---|---|---|
| Zero-Shot Transfer [33] | 4.92 | 0.57 — 0.41 | 42 |
| Domain Randomization | 1 | 0.34 — 0.23 | 19 |
| Time-Opt. Traj. + MPC [26] | 1 | 0.51 — 0.41 | 19 |
| Ours | 5.26 | 0.56 — 0.44 | 100 |

**c**

| | Approach | Ground-truth observations | | | Noisy observations | | |
|---|---|---|---|---|---|---|---|
| | | Lap time in seconds | Gate margin in meters | Completion in % | Lap time in seconds | Gate margin in meters | Completion in % |
| Idealized dyn. | Zero-Shot Transfer [33] | 4.88 | 0.63 — 0.46 | 100 | 1 | n/a — n/a | 0 |
| | Domain Randomization | 5.06 | 0.60 — 0.47 | 100 | 1 | 0.43 — 0.30 | 9 |
| | Time-Opt. Traj. + MPC [26] | 4.60 | 0.50 — 0.25 | 100 | 1 | 0.48 — 0.29 | 9 |
| | Ours | 4.88 | 0.63 — 0.46 | 100 | 5.26 | 0.56 — 0.44 | 100 |
| Realistic dyn. | Zero-Shot Transfer [33] | 1 | 0.62 — 0.62 | 4 | 1 | 0.41 — 0.21 | 9 |
| | Domain Randomization | 1 | 0.28 — 0.28 | 4 | 1 | 0.47 — 0.45 | 9 |
| | Time-Opt. Traj. + MPC [26] | 1 | n/a — n/a | 0 | 1 | 0.23 — 0.23 | 4 |
| | Ours | 5.20 | 0.51 — 0.30 | 100 | 5.42 | 0.48 — 0.23 | 100 |

**d**

| | Autonomous Drone | | | | | Vanover | | | | | Bitmatta | | | | | Schaepper | | | | |
|---|---|---|---|---|---|---|---|---|---|---|---|---|---|---|---|---|---|---|---|---|
| | Speed in m/s | Power in W | Thrust in N | Time in s | Length in m | Speed in m/s | Power in W | Thrust in N | Time in s | Length in m | Speed in m/s | Power in W | Thrust in N | Time in s | Length in m | Speed in m/s | Power in W | Thrust in N | Time in s | Length in m |
| Segment 1 | **6.52** | **548.14** | **19.80** | **1.27** | 8.27 | 5.62 | 348.91 | 14.96 | 1.42 | 7.98 | 5.93 | 365.78 | 15.87 | 1.36 | 8.07 | 4.92 | 201.06 | 10.47 | 1.59 | **7.82** |
| Segment 2 | 13.72 | 852.20 | 29.01 | **1.82** | **25.01** | 13.83 | 863.85 | **29.22** | 1.83 | 25.31 | 13.72 | 854.36 | 28.95 | 1.89 | 25.94 | 10.76 | 508.29 | 19.97 | 2.36 | 25.35 |
| Segment 3 | 13.70 | 798.69 | 27.60 | **1.96** | **26.89** | 13.86 | 853.69 | 28.98 | 2.09 | 28.91 | 13.16 | 727.36 | 25.50 | 2.23 | 29.35 | 12.66 | 636.78 | 23.47 | 2.40 | 30.34 |
| Segment 4 | 13.43 | **1047.05** | **33.69** | 1.61 | 21.62 | 12.97 | 953.47 | 31.62 | **1.60** | **20.72** | **13.47** | 1035.91 | 33.39 | 1.67 | 22.53 | 13.11 | 885.53 | 29.95 | 1.77 | 23.20 |
| Full Race | **13.11** | 866.48 | 29.16 | **17.46** | **228.85** | 12.96 | 843.50 | 28.65 | 17.96 | 232.79 | 12.89 | 822.50 | 27.96 | 18.74 | 241.52 | 11.55 | 623.53 | 22.95 | 21.16 | 244.49 |

**a**, Training hyperparameters. **b**, Comparison with baselines that are provided with the same observation noise model used by our approach. **c**, Evaluation in simulation, with idealized dynamics (top) versus realistic dynamics (bottom) and ground-truth observations (left) versus noisy observations (right). We report the fastest achieved collision-free lap time in seconds, the average and smallest gate margin of successfully passed gates and the percentage of track completed. We compare our approach with a learning-based approach that performs zero-shot transfer, with and without domain randomization during training, as well as a traditional planning and control approach[28]. **d**, Comparison of the average speed, power, thrust, time and distance travelled for each pilot during the fastest flown race. Best numbers are indicated in bold.