## [Peer Review File · Nature]

Manuscript Title: Champion-Level Performance in Drone Racing using Deep Reinforcement Learning

Reviewer Comments & Author Rebuttals

Reviewer Reports on the Initial Version:

Referees' comments:

Referee #1 (Remarks to the Author):

The authors present the first system that can fly a drone racing track as fast as human drone racing champions. Their approach, Swift, consists of first performing reinforcement learning in a simulated track. Then, test runs are performed on a real drone in a real track, and a residual observation and dynamics model is learned that captures the remaining reality gap. Swift wins the majority of individual races and also produces the fastest lap time.

I consider this work of broad public and scientific interest. As the authors say, this is the first time that a robotic system has achieved world-champion-level performance in a real-world competitive sport. The challenges involved are more complex than those in board or computer games, as this fast sport happens in the far more complex real world and requires quick decisions and acting under significant uncertainty. All the sensing and processing happens on board of the drone, which poses an enormous challenge for the employed AI methods. Although I would have wagered that drones would beat humans eventually, I did not expect this to happen already in 2022 nor based predominantly on AI methods. Hence, I would like to take this opportunity to congratulate the authors on their great achievement!

Although the article is in general very well-written and also the figures are of high quality, I do have a number of issues that should be addressed before possible publication:

1. As mentioned, onboard computing power is scarce. The authors run the Intel RealSense VIO, a deep net for gate detection, an EKF for state estimation, and a relatively small network for control. Then there is the Beta flight that translates the control commands of thrust and body rates to PWM signals to the motors. I think it is important that the authors give more information on how much compute is involved in all these steps. If the Intel VIO is closed-source, the specs of the platform can be given. Also, what is the structure and size of the deep net for gate detection? Please give more complete information on this in the current paper.

2. "These intricacies of failing perception and unmodeled dynamics are environment-, platform-, track-, and sensor-dependent. ... These residual models are integrated into the simulation and the racing policy is fine-tuned in this augmented simulation." Is the uncertainty / noise in the perception and the dynamics not also dependent on the robot's behavior? When refining the behavior in simulation, based on the residuals, could this not change the residuals? How big is this effect, and could an iterative procedure further reduce the reality gap? To be clear: Apparently, the change in behavior during refinement is not so big, since the drone successfully flies the track very fast after the refinement.

3. "At the start, Swift has a lower reaction time, taking off from the podium on average 120ms before human pilots." Since this is clearly stated here, I see no fundamental problem in it. However, I do think it would be important to comment on how to make a fair start possible. For

example, there could be a counter to help the human start in the same time, not just by feedback, but by means of predicting when the start will be. For example with: 3, 2, 1, GO!

4. The authors say about the AlphaPilot race: "The 2019 AlphaPilot autonomous drone racing competition showcased some of the best research in the field [23]. However, the first two teams still took twice as long as a professional human pilot to complete the track [24, 25].".

Two points here:

a. This statement ignores the fact that the AlphaPilot race had a very different character than the experimental setting presented here by the authors: Most importantly, whereas the current achievement took place in a hall very well-known to and controlled by the experimenters, the AlphaPilot competition took place in varying environments with different backgrounds and dynamic lighting. Moreover, compared to the current study, the training time of teams at the competition location was extremely limited (~1 hour). In AlphaPilot, the drone was provided by the organizers, so the hardware was less predictable, and interaction time with it was also severely limited. I think it is important if the authors reflect upon this, also in the article. This could be part of the discussion, in order not to break the flow in the introduction.

b. In ref [25], figure 13 shows that during the championship final, Gab707 took 8 seconds for the track, with MAVLab taking 12 seconds. On the DLR web site (<https://thedroneracingleague.com/airr/>) it says that "Gab707 guided his drone through the course in :06s. It took MavLAB :11s--proving AI drone racing is not just credible, but competitive." Although these numbers are different, neither is exactly a factor 2. In the Championship race, the winner flew against the human without possibility for changing the code for the different starting position or tuning. Moreover, the starting sequence in this case was to the disadvantage of the AI: When the starting tone was heard, the drone was allowed to start up its system. This still took ~1 second, in which the human pilot had already left the podium. This further argues for a discussion on fair starting sequences in human vs. drone races. Also, based on this and the numbers mentioned above, the comment in the introduction could at least add the word "almost" to "twice as long".

5. Methods: In general, I think the methods section does not give enough detail. If it becomes too long, some material can be relegated to an additional supplementary material document, but I do think more details are necessary to ensure understanding of the method's details and scientific reproducibility.

Smaller and bigger points that should be expanded upon:

- a. L. 370: axis of rotation: do you mean rotation direction? As in some propellers turn clockwise and others counterclockwise?
- b. What is v_{xy} ?
- c. "In simulation, we use an accurate model of this low-level controller.": What does this model look like? How are its parameters identified?
- d. L. 413 – 415: What kind of polynomial model? Can we see a plot of the function, also with e.g., predicted vs. measured voltage?
- e. L. 431 says that o has dimensionality 31. The sum involved here could be spelled out more clearly. It took me some time to figure out, because the state as shown in eq. 1 is not used here. Instead, we have 3 for position, 3 for velocity, 9 for a rotation matrix (15 so far). Then the relative pose of the gate, represented with four relative 3D positions = $15 + 12 = 27$. Then the previous action, thrust and rotation rates = $27 + 4 = 31$.
- f. The video shows sometimes two gates being detected simultaneously. Are these detections used for localization and entered as two separate observations in the Kalman filter at the same time step?
- g. L. 474: What are the activation functions?
- h. Fig. 5b: what is blue, what is orange?
- i. Fig. 5a: It is not 100% clear to me what the colored lines are. Different samplings from the

Gaussian process model? Where do we see the residual force? Is it exactly equal to the black line due to the kNN regression? Shouldn't there be multiple black lines for multiple laps?

6. "Data and materials availability: All (other) data needed to evaluate the conclusions in the paper are present in the paper or the Supplementary Materials" I can find some scripts and (psuedo-)code, but can't find any of the data. A main conclusion is that the authors beat human drone race champions. Therefore, given this statement, I would expect raw data from the races for the AI drone and the human pilots, in combination with some scripts to show them in 3D, e.g., like in Figure 4. Simultaneous onboard drone measurements would be really valuable to the community.

Small points:

- a. "Detailed analysis is provided in the supplementary material." I could not find supplementary material with text (only videos & code). Were you referring to the Methods section?
- b. In the abstract: "their aircraft's perspective" -> drone's perspective?

Referee #2 (Remarks to the Author):

Summary: The article discusses the use of deep reinforcement learning in autonomous systems, specifically in the context of first-person view drone racing. The article explains the challenges of creating autonomous systems that can compete at the level of human pilots and highlights previous attempts to solve this problem. The authors then introduce Swift, an autonomous system that can race a quadrotor at the level of human world champions while using only onboard sensors and computation. The article details the two key modules of Swift, namely the perception system and the control policy. Swift is trained using model-free on-policy deep reinforcement learning and is evaluated on a physical track against three human champions, achieving world-champion level performance.

Overall I read this paper very interestingly, here are some comments:

1. The paper employs a two-step methodology consisting of perception and control. Nevertheless, it lacks clarity on the baseline comparison as it does not specify whether the perception module remained constant while only the control policy was modified or if both perception and control were altered.
2. The paper mentions fine-tuning the original policy based on a small amount of data collected in the real world, comprising three full rollouts that correspond to about 50 seconds of flight time. The authors identify residual observations and dynamics, which are utilized for training in simulation. However, it is unclear which environment was used for fine-tuned data collection and whether idealized dynamics and ground truth observation were employed.

Referee #3 (Remarks to the Author):

The work presented shows the first instance of an autonomous mobile robot achieving world-champion-level performance in a real world competitive sport. This result is achieved by a) using deep reinforcement learning in simulation, b) enhancing the dynamics and observation simulation with real data, and c) including the uncertainty in the observation noise model. The level of detail that the authors went into, down to simulating the battery model of the quadrotor, was very impressive.

The results achieved by the proposed approach are much better than those achieved by state-of-the-art frameworks. This is the only algorithm that is able to even complete the race course, while the closest benchmark framework was only able to complete 42% of a lap. The results presented can be of interest to people working in different robotic subfields such as robotic manipulation,

where using data from the real world to improve a simulator to apply learning algorithms [1] is gaining traction. Swift, as presented in this work, shows the real potential of using real data to enhance simulation when the effects are hard to simulate. In manipulation, hard-to-simulate effects include contact dynamics.

The methodology is valid and the design choices are sound, with explanations given for these choices. The methodology is generally well presented. However, there are a few places where the text could be improved. The subsection on the battery model in line 402 is confusing. The authors state that motor speed is a function of the battery level, for a given PWM command. The authors then present how the simulator uses a model for the battery voltage. On line 413 a mapping is introduced. It is not clear what this mapping is. It seems that it is a mapping from collective thrust and battery voltage to steady-state motor speed. Moreover, it is not clear what the authors mean by steady-state motor speed since they write Ω_{des} , which has been established to be the desired motor speed. Clarification is needed as to what the mapping entails. The sentence starting on line 415 is not clear either. It seems to mean that the polynomial mapping along with the low-level controller enable the simulator to translate an action to desired motor speeds. Is this the correct interpretation? It is not clear what the low-level controller is; from context one can infer that it is the motor speed controller. Overall, it is not clear what component (Ω_{des} ?) the authors are trying to improve by using a battery model, how this simulated variable relates to the model in equation (1), and how the data is collected for the modeling described in this section. Knowing this would enable others to reproduce this work.

Regarding VIO drift estimation in line 519, naming the Kalman filter estimates as 'static drift' (line 531) and 'drift velocity' (line 532) is confusing. These names suggest that what the Kalman filter is estimating is the drift only, not the position and velocity of the drone. Based on the information in Figure 2, and the text in line 431 (which describes the robot state for the neural network) it seems that the Kalman filter is estimating the position and velocity states of the robot. Perhaps a more appropriate name would be 'estimated position'.

The authors explain their design choice of using a GP for the residual observation model. However, the choice of KN for the residual dynamics model is not presented. Additionally, the authors should clarify if they use PPO as outlined in [29], or if changes other than the reward function were made to the algorithm. Space permitting, the authors should add the reasoning behind their choice of PPO versus other reinforcement learning algorithms for Swift.

To enable other to reproduce this work, the authors should include how the Real-World experience data to improve the dynamics and the observation simulation was collected. Did an expert pilot collect the data? Was it collected using the policy trained in simulation?

The conclusions reached and data interpretation are robust. The authors should include that Swift learns to fly one specific race track. If this race track is changed, then training would have to be done again.

[1] Lim, Vincent, et al. "Real2sim2real: Self-supervised learning of physical single-step dynamic actions for planar robot casting." 2022 International Conference on Robotics and Automation (ICRA). IEEE, 2022.

Referee #4 (Remarks to the Author):

Summary: In this paper, the authors summarized the key results and technical details of the quadrotor system, Swift, developed for first-person view (FPV) drone racing. The authors pushed the limit and demonstrated that the Swift system is able to reach human-level performance. The paper is overall an enjoyable read, and the results are exciting and motivate additional research directions (e.g., robust perception approaches). I recommend this paper for publication.

Originality and significance: While the components in the Swift system are based on existing robot decision-making techniques, the thorough comparisons of competing reinforcement learning and control techniques (e.g., MPC, learning-based MPC, zero-shot transfer of model-free RL policy, and model-free RL with domain randomization), especially using imperfect state feedback, have rarely been made available in other works and are valuable for pinpointing the limitations of the state-of-the-art approaches. The authors have put in a lot of effort to push the algorithms to the limit in challenging high-speed racing tasks. As also stated in the manuscript, in the best trials of the real-world experiments, the performance of the AI-based Swift system design surpassed the top human pilots. This work demonstrated the potential of fully autonomous robots and embodied intelligence in completing tasks with human-level proficiency and competence.

Data & methodology: The overall approach is technically sound. The authors chose to leave the technical details to the supplementary material. If space permits, a few details could be useful to have in the main text (e.g., the inputs computed by the network, and the input from the human pilots, the bounds on the estimation errors from the VIO system). There are a few additional questions that would be also interesting to clarify:

1) MPC approach comparison: In the comparisons, it seems that the MPC approach effectively tracks a time-optimal trajectory, while the proposed RL approach additionally includes perception awareness in the reward function (the r^{perc} term in Equation 7 of the supplementary material). In Table 3 of the supplementary material, the authors showed that the MPC approach is not able to complete the task even when the perception and dynamics models are given. I am not sure if this comparison is completely fair for the MPC approach as neither the cost function nor the time-optimal trajectory necessarily accounts for potential perception errors resulting from the fast motions. I wonder if the failure of the MPC approach is mainly a result of the lack of perception awareness, which could be potentially mitigated by introducing additional perception constraints (e.g., similar to the role of r^{perc} in Equation 7).

2) Human pilot control: What input do the human pilots have control over (e.g., collective thrust and angular rates or altitude commands)? In the Drone Hardware Configuration Section, there is one brief statement on this, but I am not super sure as it is a little bit surprising that the human pilot also directly controls the thrust and angular rates, which are not very intuitive. It might be good to summarize the architecture of the Swift system and the case with human pilots in one high-level block diagram.

3) Perception errors: It seems that a bottleneck of the overall system designs is the accuracy of the VIO-based localization system. In particular, in the discussion section, the authors pointed out that humans are surprisingly robust to appearance changes in the scene, while the robot perception systems have poor generalization to the lighting or other environment changes. It would be interesting to include a few more figures to show the errors of the system relative to the motion capture ground truth (e.g., perception uncertainty as a function of distance to the gates).

Conclusions: The results of the paper are very promising. The authors have shown thorough comparisons in both simulations and real-world experiments. The distributions of the performance of the Swift system and human pilots are properly reported.

References: In addition to MPC, there are also other control approaches optimized for quadrotor fast flights. It would be good to mention a few of these works.

Clarity and context: The paper is overall well-written and easy to read. The abstract, introduction, and conclusion are concise and set out a good motivation for the work as well as future directions in the field. The technical details are appropriately documented and discussed. The figures, tables, and the accompanying video are generally clear and well-support the key scientific results in the work.

II. REVIEWER 1

The authors present the first system that can fly a drone racing track as fast as human drone racing champions. Their approach, Swift, consists of first performing reinforcement learning in a simulated track. Then, test runs are performed on a real drone in a real track, and a residual observation and dynamics model is learned that captures the remaining reality gap. Swift wins the majority of individual races and also produces the fastest lap time.

I consider this work of broad public and scientific interest. As the authors say, this is the first time that a robotic system has achieved world-champion-level performance in a real-world competitive sport. The challenges involved are more complex than those in board or computer games, as this fast sport happens in the far more complex real world and requires quick decisions and acting under significant uncertainty. All the sensing and processing happens on board of the drone, which poses an enormous challenge for the employed AI methods. Although I would have wagered that drones would beat humans eventually, I did not expect this to happen already in 2022 nor based predominantly on AI methods. Hence, I would like to take this opportunity to congratulate the authors on their great achievement!

Although the article is in general very well-written and also the figures are of high quality, I do have a number of issues that should be addressed before possible publication:

1. As mentioned, onboard computing power is scarce. The authors run the Intel RealSense VIO, a deep net for gate detection, an EKF for state estimation, and a relatively small network for control. Then there is the Beta flight that translates the control commands of thrust and body rates to PWM signals to the motors. I think it is important that the authors give more information on how much compute is involved in all these steps. If the Intel VIO is closed-source, the specs of the platform can be given. Also, what is the structure and size of the deep net for gate detection? Please give more complete information on this in the current paper.

We added the specs of the Intel RealSense VIO sensor, extended the methods section with a description of the gate detection network (Section *Gate Detection*), and provide runtime analyses for each main thread operating on the drone. We extended the section *Drone Hardware Configuration* with more information on the computational resources available on the autonomous drones. For completeness, we summarize the manuscript additions here as well:

- **VIO:** Visual-inertial odometry is run on an Intel RealSense T265 sensor, which uses an Intel Movidius Myriad 2 ASIC for computation. Full state estimates are provided at a frequency of 100 Hz.
- **Gate detection network:** The gate detection network is represented by a 6-level U-Net architecture that operates on grayscale frames of size 384x384. One forward pass through the network takes 40 ms on an NVIDIA Jetson TX2.
- **Kalman filter:** The Kalman filter measurement update requires less than 1 ms, even when four gates are detected simultaneously (we never detected more than four gates simultaneously in a single frame).
- **Policy network:** Policy inference is performed on the CPU and takes 8 ms on an NVIDIA Jetson TX2.
- **Flight controller:** The open-source flight controller software Betaflight is running on an STM32F7 microprocessor chip with a clock frequency of 216 MHz.

2. “These intricacies of failing perception and unmodeled dynamics are environment-, platform-, track-, and sensor-dependent. . . . These residual models are integrated into the simulation and the racing policy is fine-tuned in this augmented simulation.” Is the uncertainty / noise in the perception and the dynamics not also dependent on the robot’s behavior? When refining the behavior in simulation, based on the residuals, could this not change the residuals? How big is this effect, and could an iterative procedure further reduce the reality gap? To be clear: Apparently, the change in behavior during refinement is not so big, since the drone successfully flies the track very fast after the refinement.

We conducted an investigation to comprehend the extent of variations in behavior and residuals across iterations. The findings of our analysis reveal that subsequent finetuning operations result in negligible enhancements in performance and alterations in behavior (Fig. 1). Notably, residual components capture the predominant source of noise and uncertainty in perception after a single iteration (Fig. 2). We hypothesize this to be the case since our initial models possess a high degree of accuracy and the environmental changes between iterations are comparatively minor. We added this analysis to the supplementary material in section *Finetuning for Multiple Iterations*.

In the following, we provide more details on this investigation. We start by enumerating the finetuning steps to provide the necessary notation:

- 1) Train policy-0 in simulation.
- 2) Deploy policy-0 in the real world. The policy operates on ground-truth data from a motion capture system.
- 3) Identify residuals observed by policy-0 in the real world.
- 4) Train policy-1 by finetuning policy-0 on the identified residuals.
- 5) Deploy policy-1 in the real world. The policy operates only on onboard sensory measurements.
- 6) Identify residuals observed by policy-1 in the real world.
- 7) Train policy-2 by finetuning policy-1 on the identified residuals.

We compare performance of policy-1 and policy-2 in simulation after finetuning on their respective residuals. The results are illustrated in Fig. 1. We observe that the difference in distance from gate centers, which is a metric for the safety of the policy, is $0.09 \pm 0.08\text{m}$. Furthermore, the difference in the time taken to complete a single lap is $0.02 \pm 0.02\text{s}$. Note that this laptime difference is substantially smaller than the difference between the single-lap completion times of Swift and the human pilots (0.16s).

Fig. 1: Rollout comparison after fine-tuning the policy for one iteration (blue) and two iterations (orange).

Fig. 2: Difference between residuals observed by policy-0 and policy-1 in real-world deployments.

Fig. 3: Perception residuals observed by policy-0 in real world deployments.

To investigate why subsequent finetuning iterations have only a marginal effect, we evaluate the difference in residuals observed by policy-0 and policy-1 in the real world. Specifically, Fig. 2 illustrates the difference in residuals between policy-0 and policy-1. We observe that the mean of such difference is much smaller than the actual residuals observed by policy-0 (Fig. 3). This indicates that residuals do not change significantly between iterations, thereby having only a small impact on

behaviour.

3. “At the start, Swift has a lower reaction time, taking off from the podium on average 120ms before human pilots.” Since this is clearly stated here, I see no fundamental problem in it. However, I do think it would be important to comment on how to make a fair start possible. For example, there could be a counter to help the human start in the same time, not just by feedback, but by means of predicting when the start will be. For example with: 3, 2, 1, GO!

We followed official drone racing rules, which require randomized starting sequences [1]. Note also that the best race time of our approach is 0.5 seconds faster than the best human pilot, which is larger than the 120 ms starting latency of the human pilot.

[1] FAI Drone Sports: Documents and Rules [Link], [PDF]

4. The authors say about the AlphaPilot race: “The 2019 AlphaPilot autonomous drone racing competition showcased some of the best research in the field [23]. However, the first two teams still took twice as long as a professional human pilot to complete the track [24, 25].”

Two points here:

- (a) This statement ignores the fact that the AlphaPilot race had a very different character than the experimental setting presented here by the authors: Most importantly, whereas the current achievement took place in a hall very well-known to and controlled by the experimenters, the AlphaPilot competition took place in varying environments with different backgrounds and dynamic lighting. Moreover, compared to the current study, the training time of teams at the competition location was extremely limited (1 hour). In AlphaPilot, the drone was provided by the organizers, so the hardware was less predictable, and interaction time with it was also severely limited. I think it is important if the authors reflect upon this, also in the article. This could be part of the discussion, in order not to break the flow in the introduction.
- (b) In ref [25], figure 13 shows that during the championship final, Gab707 took 8 seconds for the track, with MAVLab taking 12 seconds. On the DLR web site (<https://thedroneracingleague.com/airr/>) it says that “Gab707 guided his drone through the course in :06s. It took MavLAB :11s—proving AI drone racing is not just credible, but competitive.” Although these numbers are different, neither is exactly a factor 2. In the Championship race, the winner flew against the human without possibility for changing the code for the different starting position or tuning. Moreover, the starting sequence in this case was to the disadvantage of the AI: When the starting tone was heard, the drone was allowed to start up its system. This still took 1 second, in which the human pilot had already left the podium. This further argues for a discussion on fair starting sequences in human vs. drone races. Also, based on this and the numbers mentioned above, the comment in the introduction could at least add the word “almost” to “twice as long”.

-
- (a) We agree with the reviewer that the contestants in the AlphaPilot challenge only had limited access to the track layout and environment before racing. We updated the manuscript (Section *Discussion*) to emphasize this fact better.
 - (b) We implemented the reviewer’s suggestion and updated the mentioned sentence in the manuscript by adding the word ‘almost’.

5. Methods: In general, I think the methods section does not give enough detail. If it becomes too long, some material can be relegated to an additional supplementary material document, but I do think more details are necessary to ensure understanding of the method’s details and scientific reproducibility.

We extended the methods section by (i) adding a description of the gate detection network, (ii) providing additional information on the battery model, and (iii) by providing further clarifications on the VIO drift estimation and the simulation results.

Smaller and bigger points that should be expanded upon:

- (a) L. 370: axis of rotation: do you mean rotation direction? As in some propellers turn clockwise and others counterclockwise?
- (b) What is v_{xy} ?
- (c) “In simulation, we use an accurate model of this low-level controller.”: What does this model look like? How are its parameters identified?
- (d) L. 413 – 415: What kind of polynomial model? Can we see a plot of the function, also with e.g., predicted vs. measured voltage?
- (e) L. 431 says that o has dimensionality 31. The sum involved here could be spelled out more clearly. It took me some time to figure out, because the state as shown in eq. 1 is not used here. Instead, we have 3 for position, 3 for velocity, 9 for a rotation matrix (15 so far). Then the relative pose of the gate, represented with four relative 3D positions = $15 + 12 = 27$. Then the previous action, thrust and rotation rates = $27 + 4 = 31$.
- (f) The video shows sometimes two gates being detected simultaneously. Are these detections used for localization and entered as two separate observations in the Kalman filter at the same time step?

- (g) L. 474: What are the activation functions?
 - (h) Fig. 5b: what is blue, what is orange?
 - (i) Fig. 5a: It is not 100% clear to me what the colored lines are. Different samplings from the Gaussian process model? Where do we see the residual force? Is it exactly equal to the black line due to the kNN regression? Shouldn't there be multiple black lines for multiple laps?
-

- (a) The motor reaction torque generated from all propellers is given by the sum of each reaction torque generated by each individual propeller. Each individual reaction torque can be described as the product of the motor/propeller inertia with the change in angular velocity, which is a vector in 3D space described parallel to the rotation axis of the rotor.
 - (b) We have clarified the notation used for the residual aerodynamic model. Specifically, v_{xy} refers to the bodyframe xy-velocity, $v_{xy} = \sqrt{v_x^2 + v_y^2}$.
 - (c) To reduce the sim-to-real gap the control architecture used in simulation closely follows the original Betaflight control design. In particular, we use a PID controller with gains identified from the detailed onboard logs that reveal the PID controller's internal state. As the Betaflight controller is optimized for human pilots, it shows some peculiarities which we have detailed in the manuscript.
 - (d) The lines in question describe a model that maps a thrust command and a voltage to a given motor speed. As such, a plot that shows predicted vs. measured voltage would not illustrate the model that is used. Such a plot can be found in Reference [42] (Fig. 6). We have changed the text to be more clear and highlighted that *given* a battery voltage and a command, the polynomial fit outputs the steady-state motor speed. We also added the terms included in the polynomial fit.
 - (e) The sum described by the reviewer is correct. We added a more explicit derivation of the dimensionality of the input tensor to the manuscript in Section *Observations, actions, and rewards*.
 - (f) Yes, if multiple gates are observed, the measurements are stacked and processed simultaneously by the Kalman filter. We updated the manuscript (Section *VIO Drift Estimation*) to clarify this.
 - (g) We use LeakyReLU activation functions with a slope of 0.2 in the negative halfplane. The manuscript is updated accordingly in Section *Training Details*.
 - (h) Blue: ground truth pose from simulator. Orange: ground truth pose + simulated observation residual. We updated the manuscript by extending the caption of Fig. 5b.
 - (i) The colored lines represent different sampled realizations of the Gaussian process model. The residual force is plotted in the bottom row. As the residual force component is modeled by a deterministic model, it is represented by a single line for the race. Each plot spans one entire race, i.e. 3 laps. We updated the caption of Fig. 5a.
-

6. "Data and materials availability: All (other) data needed to evaluate the conclusions in the paper are present in the paper or the Supplementary Materials" I can find some scripts and (psuedo-)code, but can't find any of the data. A main conclusion is that the authors beat human drone race champions. Therefore, given this statement, I would expect raw data from the races for the AI drone and the human pilots, in combination with some scripts to show them in 3D, e.g., like in Figure 4. Simultaneous onboard drone measurements would be really valuable to the community.

We added all the collected race data to a public repository, together with a set of scripts to visualize the data and compute lap times. The race data and scripts can be found at this [Link].

Small points:

- (a) "Detailed analysis is provided in the supplementary material." I could not find supplementary material with text (only videos & code). Were you referring to the Methods section?
 - (b) In the abstract: "their aircraft's perspective" → drone's perspective?
-
- (a) Yes, we were referring to the methods section.
 - (b) We updated the abstract and implemented the reviewer's suggestion.

III. REVIEWER 2

Summary: The article discusses the use of deep reinforcement learning in autonomous systems, specifically in the context of first-person view drone racing. The article explains the challenges of creating autonomous systems that can compete at the level of human pilots and highlights previous attempts to solve this problem. The authors then introduce Swift, an autonomous system that can race a quadrotor at the level of human world champions while using only onboard sensors and computation. The article details the two key modules of Swift, namely the perception system and the control policy. Swift is trained using model-free on-policy deep reinforcement learning and is evaluated on a physical track against three human champions, achieving world-champion level performance.

Overall I read this paper very interestingly, here are some comments:

1. The paper employs a two-step methodology consisting of perception and control. Nevertheless, it lacks clarity on the baseline comparison as it does not specify whether the perception module remained constant while only the control policy was modified or if both perception and control were altered.

The perception itself remained constant during the finetuning process, while the control policy was altered to account for changing perception. Throughout the ablation studies in simulation, perception is kept constant: we simulate observation errors by sampling one realization from the residual observation model that is identified from real-world data. We, therefore, do not simulate the onboard camera itself. We updated the manuscript in Section *Residual Model Identification* and also in Section *Simulation Results* to clarify this aspect.

2. The paper mentions fine-tuning the original policy based on a small amount of data collected in the real world, comprising three full rollouts that correspond to about 50 seconds of flight time. The authors identify residual observations and dynamics, which are utilized for training in simulation. However, it is unclear which environment was used for fine-tuned data collection and whether idealized dynamics and ground truth observation were employed.

In the finetuning stage, we use a simulator with a dynamics and perception model optimized using real-world experience. Specifically, these optimized models are defined by a residual learned with real-world data. This residual is added to the idealized dynamics and ground-truth observations. We clarified this point in the section *Residual Model Identification* of the appendix.

The real-world data used for fine-tuning is collected by deploying a policy trained in simulation in the real world using a motion capture system. This system allows collecting both accurate ground-truth state estimation as well as data from the onboard perception system while the drone is flying autonomously through the race track. We updated the manuscript in Section *The Swift System* to clarify that the drone is racing autonomously during the initial data collection phase.

IV. REVIEWER 3

The work presented shows the first instance of an autonomous mobile robot achieving world-champion-level performance in a real-world competitive sport. This result is achieved by a) using deep reinforcement learning in simulation, b) enhancing the dynamics and observation simulation with real data, and c) including the uncertainty in the observation noise model. The level of detail that the authors went into, down to simulating the battery model of the quadrotor, was very impressive.

The results achieved by the proposed approach are much better than those achieved by state-of-the-art frameworks. This is the only algorithm that is able to even complete the race course, while the closest benchmark framework was only able to complete 42% of a lap. The results presented can be of interest to people working in different robotic subfields such as robotic manipulation, where using data from the real world to improve a simulator to apply learning algorithms [1] is gaining traction. Swift, as presented in this work, shows the real potential of using real data to enhance simulation when the effects are hard to simulate. In manipulation, hard-to-simulate effects include contact dynamics.

The methodology is valid and the design choices are sound, with explanations given for these choices. The methodology is generally well presented. However, there are a few places where the text could be improved. The subsection on the battery model in line 402 is confusing. The authors state that motor speed is a function of the battery level, for a given PWM command. The authors then present how the simulator uses a model for the battery voltage. On line 413 a mapping is introduced. It is not clear what this mapping is. It seems that it is a mapping from collective thrust and battery voltage to steady-state motor speed. Moreover, it is not clear what the authors mean by steady-state motor speed since they write Ω_{des} , which has been established to be the desired motor speed. Clarification is needed as to what the mapping entails. The sentence starting on line 415 is not clear either. It seems to mean that the polynomial mapping along with the low-level controller enable the simulator to translate an action to desired motor speeds. Is this the correct interpretation? It is not clear what the low-level controller is; from context one can infer that it is the motor speed controller. Overall, it is not clear what component (Ω_{des} ?) the authors are trying to improve by using a battery model, how this simulated variable relates to the model in equation (1), and how the data is collected for the modeling described in this section. Knowing this would enable others to reproduce this work.

The detailed observation by the reviewer was correct, and the notation we used was oversimplified, which led to the confusion. We have extended the corresponding section and adapted the notation to be precise. To address the points mentioned:

- We used the term ‘low-level controller’ for a system that consists of both a PID controller that maps the collective thrust and bodyrate commands to individual motor commands and an ESC (electronic speed controller) which translates the motor commands to analog voltage signals for the motors. We have clarified this in the manuscript by clearly separating the low-level controller from the ESC which is now mentioned as a separate component.
- The desired motor speed Ω_{des} is related to the thrust command, but since neither ESC nor the low-level controller perform closed-loop control on motor thrusts or speeds, we do not control motor speeds directly. Therefore, given a constant motor command, the actual motor speed which the propeller accelerates to will depend on the battery voltage. To highlight this, we have renamed the desired motor speed to ‘steady-state motor speed’ everywhere.
- Since we do not control the steady-state motor speed directly, our simulated low-level controller and ESC need to calculate how fast the motors would spin given the battery voltage. This requires an accurate simulation of the battery voltage. Based on this, the polynomial mapping maps a given individual motor command and battery voltage to a steady-state motor speed.
- The parameters are identified from Betaflight logs which contain information about the current battery voltage, the motor commands, and motor speeds.

Regarding VIO drift estimation in line 519, naming the Kalman filter estimates as ‘static drift’ (line 531) and ‘drift velocity’ (line 532) is confusing. These names suggest that what the Kalman filter is estimating is the drift only, not the position and velocity of the drone. Based on the information in Figure 2, and the text in line 431 (which describes the robot state for the neural network) it seems that the Kalman filter is estimating the position and velocity states of the robot. Perhaps a more appropriate name would be ‘estimated position’.

The Kalman filter used in this work estimates the positional drift of the VIO system as well as its derivative. Note that the Kalman filter does not directly estimate the full position and velocity of the drone, but only the translational error states of the VIO system. We describe the state of the Kalman filter in the second paragraph of Section *VIO Drift Estimation*, where we renamed *static drift* to *translational drift*.

The authors explain their design choice of using a GP for the residual observation model. However, the choice of KNN for the residual dynamics model is not presented. Additionally, the authors should clarify if they use PPO as outlined in [29], or if changes other than the reward function were made to the algorithm. Space permitting, the authors should add the reasoning behind their choice of PPO versus other reinforcement learning algorithms for Swift.

- We empirically noticed that the drift of the visual odometry pipeline is stochastic: even a tiny difference in the initialization procedure can lead to large differences in drift throughout the track (Fig. 5, top three rows). We used Gaussian Processes to capture the stochasticity of the drift. Conversely, since we operate in an indoor arena, we expect residual dynamics to be much more deterministic for a given drone. We confirm this hypothesis experimentally (Fig. 5, bottom row). For this reason, a deterministic model is sufficient for the task. We empirically found that the simplest fitting model (KNN) balances data efficiency and performance well.
- Our implementation uses PPO as outlined in [29] without additional changes to the algorithm. Our approach is orthogonal to the choice of the optimization algorithm. We used PPO because of its proven ability to learn policies that perform well in the real world [1, 2, 3].

[1] Lee, J., Hwangbo, J., Wellhausen, L., Koltun, V. and Hutter, M., 2020. Learning quadrupedal locomotion over challenging terrain. *Science robotics*, 5(47).

[2] Kumar, A., Fu, Z., Pathak, D. and Malik, J., 2021. Rma: Rapid motor adaptation for legged robots. *Robotics: Science and Systems*.

[3] Radosavovic, I., Xiao, T., James, S., Abbeel, P., Malik, J. and Darrell, T., 2023, March. Real-world robot learning with masked visual pre-training. In *Conference on Robot Learning*.

To enable other to reproduce this work, the authors should include how the Real-World experience data to improve the dynamics and the observation simulation was collected. Did an expert pilot collect the data? Was it collected using the policy trained in simulation?

The real-world experience data to improve the dynamics and perception models was collected by running the policy trained in simulation on accurate state estimates obtained from a motion capture system. We updated the manuscript in Section *The Swift System* to clarify that the drone is racing autonomously during the initial data collection phase.

The conclusions reached and data interpretation are robust. The authors should include that Swift learns to fly one specific race track. If this race track is changed, then training would have to be done again.

[1] Lim, Vincent, et al. "Real2sim2real: Self-supervised learning of physical single-step dynamic actions for planar robot casting." 2022 International Conference on Robotics and Automation (ICRA). IEEE, 2022.

Training for a specific race track layout is an integral part of drone racing (as well as other sports such as car racing). Human pilots also train on the specific track before racing. We updated the manuscript in Section *Discussion* to emphasize that our approach is trained for a specific track layout and requires retraining when the track layout changes.

V. REVIEWER 4

Summary: In this paper, the authors summarized the key results and technical details of the quadrotor system, Swift, developed for first-person view (FPV) drone racing. The authors pushed the limit and demonstrated that the Swift system is able to reach human-level performance. The paper is overall an enjoyable read, and the results are exciting and motivate additional research directions (e.g., robust perception approaches). I recommend this paper for publication.

Originality and significance: While the components in the Swift system are based on existing robot decision-making techniques, the thorough comparisons of competing reinforcement learning and control techniques (e.g., MPC, learning-based MPC, zero-shot transfer of model-free RL policy, and model-free RL with domain randomization), especially using imperfect state feedback, have rarely made available in other works and are valuable for pinpointing the limitations of the state-of-the-art approaches. The authors have put in a lot of effort to push the algorithms to the limit in challenging high-speed racing tasks. As also stated in the manuscript, in the best trials of the real-world experiments, the performance of the AI-based Swift system design surpassed the top human pilots. This work demonstrated the potential of fully autonomous robots and embodied intelligence in completing tasks with human-level proficiency and competence.

Data & methodology: The overall approach is technically sound. The authors chose to leave the technical details to the supplementary material. If space permits, a few details could be useful to have in the main text (e.g., the inputs computed by the network, and the input from the human pilots, the bounds on the estimation errors from the VIO system).

Due to space constraints, we kept the details regarding network inputs and the VIO system in the methods section.

There are a few additional questions that would be also interesting to clarify:

1) MPC approach comparison: In the comparisons, it seems that the MPC approach effectively tracks a time-optimal trajectory, while the proposed RL approach additionally includes perception awareness in the reward function (the r^{perc} term in Equation 7 of the supplementary material). In Table 3 of the supplementary material, the authors showed that the MPC approach is not able to complete the task even when the perception and dynamics models are given. I am not sure if this comparison is completely fair for the MPC approach as neither the cost function nor the time-optimal trajectory necessarily accounts for potential perception errors resulting from the fast motions. I wonder if the failure of the MPC approach is mainly a result of the lack of perception awareness, which could be potentially mitigated by introducing additional perception constraints (e.g., similar to the role of r^{perc} in Equation 7).

The findings presented in Table 2 reveal that, in spite of having access to a ground-truth state, the MPC algorithm is unable to successfully complete the track, as evidenced by the bottom-left block of the table. Notably, this represents an upper limit on performance in scenarios involving noisy observations, regardless of any additional perception constraints. To enable successful completion of the lap, one approach is to track a slower trajectory, as demonstrated in prior research [1]. However, it is important to note that this approach results in lower lap times during real-world testing than our fastest policy. Indeed, we achieve an average lap time of 5.52 seconds with our RL controller, compared to the MPC's lap time of 6.05 seconds, despite the MPC being provided with ground-truth state from a Vicon system.

[1] Foehn, P., Romero, A. and Scaramuzza, D., 2021. Time-optimal planning for quadrotor waypoint flight. *Science Robotics*, 6(56), p.eabh1221.

2) Human pilot control: What input do the human pilots have control over (e.g., collective thrust and angular rates or altitude commands)? In the Drone Hardware Configuration Section, there is one brief statement on this, but I am not super sure as it is a little bit surprising that the human pilot also directly controls the thrust and angular rates, which are not very intuitive. It might be good to summarize the architecture of the Swift system and the case with human pilots in one high-level block diagram.

The human pilots directly control collective thrust and bodyrates. This control modality is typically only used by experienced pilots. We extended the caption of the block diagram in Fig. 2 in the main manuscript to clarify that the control modality of the Swift system and the human pilots is the same.

3) Perception errors: It seems that a bottleneck of the overall system designs is the accuracy of the VIO-based localization system. In particular, in the discussion section, the authors pointed out that humans are surprisingly robust to appearance changes in the scene, while the robot perception systems have poor generalization to the lighting or other environment changes. It would be interesting to include a few more figures to show the errors of the system relative to the motion capture ground truth (e.g., perception uncertainty as a function of distance to the gates).

Fig. 5a in the manuscript depicts the error of the onboard perception system relative to the motion capture ground truth for a full race (i.e., 3 laps). We have now added this clarification to the caption.

Fig. 5b in the manuscript shows the simulated positional errors that are generated by sampling one realization of the observation residuals identified from real data.

Fig. 4 (here below) shows the positional error observed during a vision-based race. The dependency of the estimation system on the gate detections is well visible when the drone coming from the region indicated by the red circle approaches

the top-right gate indicated by the red arrow: As soon as the drone is close enough to the gate (approx. at [2.5/0.0] in Fig. 4), the estimated position (solid line) rapidly becomes much more accurate.

Fig. 4: Illustration of the positional error observed during a vision-based race. Solid lines depict the estimated position generated by fusing VIO with the gate detections. Dashed lines are the ground-truth measurements from the motion tracking system.

Conclusions: The results of the paper are very promising. The authors have shown thorough comparisons in both simulations and real-world experiments. The distributions of the performance of the Swift system and human pilots are properly reported.

References: In addition to MPC, there are also other control approaches optimized for quadrotor fast flights. It would be good to mention a few of these works.

We agree with the reviewer that various approaches exist for quadrotor fast flight. We added the following citations in the main text [1,2], but we are open to other suggestions from the reviewer.

[1] Ryou, G., Tal, E. and Karaman, S., 2021. Multi-fidelity black-box optimization for time-optimal quadrotor maneuvers. *The International Journal of Robotics Research*, 40(12-14), pp.1352-1369.

[2] Pham, H. and Pham, Q.C., 2018. A new approach to time-optimal path parameterization based on reachability analysis. *IEEE Transactions on Robotics*, 34(3), pp.645-659.

Clarity and context: The paper is overall well-written and easy to read. The abstract, introduction, and conclusion are concise and set out a good motivation for the work as well as future directions in the field. The technical details are appropriately documented and discussed. The figures, tables, and the accompanying video are generally clear and well-support the key scientific results in the work.

Reviewer Reports on the First Revision:

Referees' comments:

Referee #1 (Remarks to the Author):

The authors have addressed my comments very well. The methodology is much clearer now, facilitating scientific reproduction of the results, and the flight data (positions and angles over time) can at least allow verification of the main conclusions. Moreover, I appreciate the extra work and analysis on iterative model refinement - very interesting.

Concerning the starting procedure: I indeed believe that the authors followed existing rules, but these have been made for human-against-human drone races. What I asked was some reflection on a better procedure for human-against-drone races. I now leave this up to the authors, whether they take this on board towards a final version of the article.

Another tiny remark: According to the authors' definition, v_{xy} is the *speed* in the horizontal plane. "Velocity" implies directionality. It may help to insert the definition that the authors showed me in the rebuttal in the methodology part of the article (around line 399).

Referee #3 (Remarks to the Author):

Thank you to the authors for addressing the comments. I am happy with how the comments were addressed.

There is one last minor comment. The Kalman filter estimates the drift of the position estimates produced by VIO. Is the position drift from the Kalman Filter added/subtracted from the position estimates produced by VIO? Are the position drift and estimated position related in a different manner? This minor clarification would help understand the observation of the policy network (stated in line 465 that it takes in the estimated position of the robot state).

Referee #4 (Remarks to the Author):

I would like to thank the authors for addressing my comments from the first round of reviews. I only have one further comment regarding the RL and MPC comparison. In the responses, the authors mentioned that the MPC could not complete the task for the case with realistic dynamics and ground truth feedback. This failure can be mostly attributed to model mismatch. Based on the discussion in the "Residual Model Identification" subsection, the RL agents are trained in the more realistic setting with residual dynamics and residual observation models, while the MPC approach was designed based on the idealized model but tested in the realistic setting. I am still not very convinced that this could be considered a fair comparison. After all, the two types of uncertainties (observation noise and dynamics error) can be detrimental to the model-based approaches. Still, there are robust approaches (e.g., robust MPC) to account for these errors in either the cost function or the constraints; these variants are not mentioned in the discussion. From the top left block in Table 2, we can see that the MPC approach is more optimal in the ideal setting, while from the other blocks, the MPC approach has poorer generalization with respect to observation noise and dynamics errors. This trend looks more like an inherent optimality and robustness tradeoff between the RL and the MPC approaches (which can be dependent on the parameter/hyperparameter choices of the algorithms). To help readers better understand the

context of the comparison, I think it would be useful to clarify how the parameters/hyperparameters were chosen for the compared algorithms.

Author Rebuttals to First Revision:

Reviewer Feedback

Referee #1 (Remarks to the Author):

The authors have addressed my comments very well. The methodology is much clearer now, facilitating scientific reproduction of the results, and the flight data (positions and angles over time) can at least allow verification of the main conclusions. Moreover, I appreciate the extra work and analysis on iterative model refinement - very interesting.

Concerning the starting procedure: I indeed believe that the authors followed existing rules, but these have been made for human-against-human drone races. What I asked was some reflection on a better procedure for human-against-drone races. I now leave this up to the authors, whether they take this on board towards a final version of the article.

Another tiny remark: According to the authors' definition, v_{xy} is the *speed* in the horizontal plane. "Velocity" implies directionality. It may help to insert the definition that the authors showed me in the rebuttal in the methodology part of the article (around line 399).

We thank the reviewer for the valuable feedback. We updated the manuscript as suggested by clarifying that \$v_{xy}\$ denotes the platform's speed in the inertial xy plane.

Referee #3 (Remarks to the Author):

Thank you to the authors for addressing the comments. I am happy with how the comments were addressed.

There is one last minor comment. The Kalman filter estimates the drift of the position estimates produced by VIO. Is the position drift from the Kalman Filter added/subtracted from the position estimates produced by VIO? Are the position drift and estimated position related in a different manner? This minor clarification would help understand the observation of the policy network (stated in line 465 that it takes in the estimated position of the robot state).

We thank the reviewer for the valuable feedback. The reviewer is correct, we subtract the drift estimated by the Kalman filter from the state estimates produced by VIO. We updated the manuscript to clarify this aspect.

Referee #4 (Remarks to the Author):

I would like to thank the authors for addressing my comments from the first round of reviews. I only have one further comment regarding the RL and MPC comparison. In the responses, the authors mentioned that the MPC could not complete the task for the case with realistic

dynamics and ground truth feedback. This failure can be mostly attributed to model mismatch. Based on the discussion in the “Residual Model Identification” subsection, the RL agents are trained in the more realistic setting with residual dynamics and residual observation models, while the MPC approach was designed based on the idealized model but tested in the realistic setting. I am still not very convinced that this could be considered a fair comparison. After all, the two types of uncertainties (observation noise and dynamics error) can be detrimental to the model-based approaches. Still, there are robust approaches (e.g., robust MPC) to account for these errors in either the cost function or the constraints; these variants are not mentioned in the discussion. From the top left block in Table 2, we can see that the MPC approach is more optimal in the ideal setting, while from the other blocks, the MPC approach has poorer generalization with respect to observation noise and dynamics errors. This trend looks more like an inherent optimality and robustness tradeoff between the RL and the MPC approaches (which can be dependent on the parameter/hyperparameter choices of the algorithms). To help readers better understand the context of the comparison, I think it would be useful to clarify how the parameters/hyperparameters were chosen for the compared algorithms.

We thank the reviewer for the constructive feedback. The failure of MPC can be partially attributed to model mismatch, which could potentially be alleviated by robust control methods. However, we would like to point the reader to follow-up work from our lab that delves into the reasons why RL outperforms optimal control (OC), including MPC, for autonomous racing [1]. The core findings, quoting [1], are: *“Our study indicates that the fundamental advantage of RL over OC is not that it optimizes its objective better but that it optimizes a better objective: OC decomposes the problem into planning and control with an explicit intermediate representation, such as a trajectory, that serves as an interface. This decomposition limits the range of behaviors that can be expressed by the controller, leading to inferior control performance when facing unmodeled effects. In contrast, RL can directly optimize a task-level objective and can leverage domain randomization to cope with model uncertainty, allowing the discovery of more robust control responses.”*

[1] Song, Y., Romero, A., Müller, M., Koltun, V. and Scaramuzza, D., Reaching the Limit in Autonomous Racing: Optimal Control versus Reinforcement Learning. Science Robotics, Sep., 2023.